# See What You Are Told: Visual Attention Sink in Large Multimodal Models

**Seil Kang**[*]  **Jinyeong Kim**[*]  **Junhyeok Kim**   **Seong Jae Hwang**[†]
Yonsei University
`{seil,jinyeong1324,timespt,seongjae}@yonsei.ac.kr`

## Abstract

Large multimodal models (LMMs) "see" images by leveraging the attention mechanism between text and visual tokens in the transformer decoder. Ideally, these models should focus on key visual information relevant to the text token. However, recent findings indicate that LMMs have an extraordinary tendency to consistently allocate high attention weights to specific visual tokens, even when these tokens are irrelevant to the corresponding text. In this study, we investigate the property behind the appearance of these irrelevant visual tokens and examine their characteristics. Our findings show that this behavior arises due to the massive activation of certain hidden state dimensions, which resembles the attention sink found in language models. Hence, we refer to this phenomenon as the *visual attention sink*. In particular, our analysis reveals that removing the irrelevant visual sink tokens does not impact model performance, despite receiving high attention weights. Consequently, we recycle the attention to these tokens as surplus resources, redistributing the attention budget to enhance focus on the image. To achieve this, we introduce Visual Attention Redistribution (VAR), a method that redistributes attention in image-centric heads, which we identify as innately focusing on visual information. VAR can be seamlessly applied across different LMMs to improve performance on a wide range of tasks, including general vision-language tasks, visual hallucination tasks, and vision-centric tasks, all without the need for additional training, models, or inference steps. Experimental results demonstrate that VAR enables LMMs to process visual information more effectively by adjusting their internal attention mechanisms, offering a new direction to enhancing the multimodal capabilities of LMMs.

## 1 Introduction

Large multimodal models (LMMs) have been actively expanding the capabilities of large language models to multimodal tasks (Liu et al., 2024c;a;b; Li et al., 2023b; Bai et al., 2023). In particular, the LMMs leverage a pre-trained visual encoder (Radford et al., 2021) to process image data and the transformer decoder of a large language model to generate text responses (OpenAI, 2023; Touvron et al., 2023; Yang et al., 2024). This straightforward yet powerful architecture has proven highly effective in utilizing visual information from images for vision-language tasks such as visual question answering, image captioning, and visual reasoning (Peng et al., 2023; Alayrac et al., 2022; Tsimpoukelli et al., 2021).

To incorporate visual information into text responses, LMMs rely on the attention mechanism (Vaswani et al., 2017) within the transformer decoder. Specifically, when processing multimodal inputs, the attention weights between visual and text tokens determine how much each text token focuses on the corresponding visual information. For instance, as illustrated in the top-left corner of Fig. 1, when the text token is '`bird`', the model concentrates on visual tokens associated with the bird in the image. Intuitively, LMMs should primarily attend to the visual tokens that are relevant to each text token.

---

[*]Equal contribution.
[†]Corresponding author.

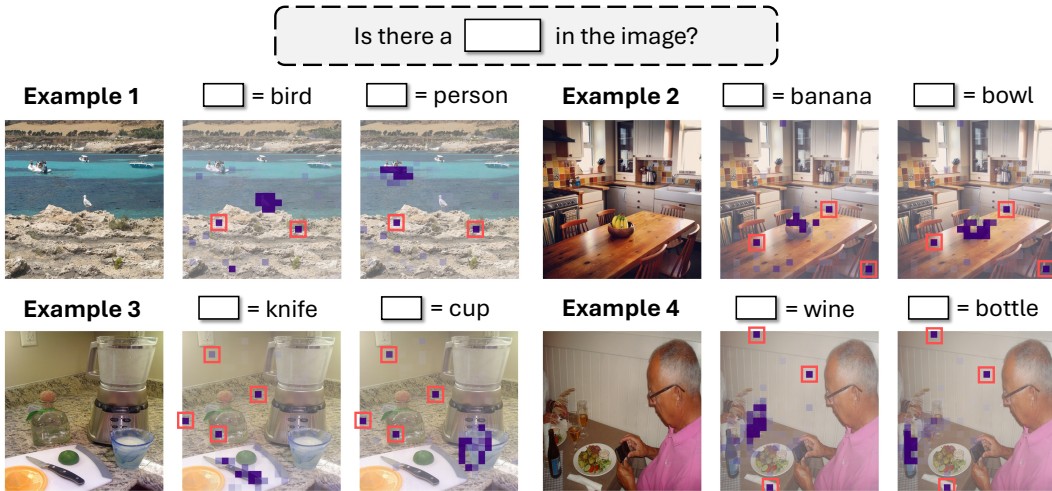

Figure 1: **Visual attention maps of LLaVA-1.5-7B between specified text tokens and visual tokens.** Attention map visualizes where the model "see" when processing the text token. The model is expected to focus only on the visual tokens related to each text token. However, the model also attends to irrelevant visual tokens ( red boxes ) that are unrelated to the corresponding text token. Although we visualize the attention maps only for a few specified text tokens, these irrelevant tokens consistently occur in fixed locations across the entire text tokens, including the instructions and the generated responses (see Fig. 13 in Appendix for more examples).

However, in practice, not all attention is directed toward the relevant visual tokens. As shown in Fig. 1, the model also allocates high attention weights to visual tokens which are unrelated to the corresponding text. This phenomenon is widespread in LMMs (Woo et al., 2024; An et al., 2024), and a notable pattern emerges where irrelevant visual tokens consistently appear in fixed locations across different text tokens. For example, in each case illustrated in Fig. 1, irrelevant visual tokens (highlighted in red boxes) consistently occupy the same positions regardless of the text tokens, indicating an underlying pattern. The cause and meaning of this phenomenon remain open questions, motivating this study.

In this work, we explore the underlying property and the characteristics of the irrelevant visual tokens. We find that these tokens in the visual attention map arise from the massive activation of specific dimensions in the hidden states. This mechanism is analogous to the attention sink observed in language models, where the model consistently assigns large attention weights to tokens with limited semantic meaning (e.g., 'BOS', '.', '\n', etc.) (Xiao et al., 2023; Sun et al., 2024a). Irrelevant visual tokens can be identified by the extreme magnitudes in a few specific dimensions, which we refer to as *visual sink tokens*, as they also have limited semantic information from the image. Furthermore, we demonstrate that removing these visual sink tokens does not significantly impact the quality of the model's response, despite the model assigning high attention weights to them.

Based on these experiments, we propose that the attention weights assigned to sink tokens can be recycled as an "attention budget". Since recent studies have reported that attention allocated to images is often insufficient compared to that given to text (Chen et al., 2024; Liu et al., 2024d), we redistribute the excess attention from sink tokens to image. Also, considering each attention head serves a distinct function (Zheng et al., 2024), we identify the heads that are primarily responsible for focusing on visual information, namely, image-centric heads, based on the presence of visual attention sinks. Finally, we introduce Visual Attention Redistribution (VAR), a two-step method: first, selecting the image-centric heads, and second, redistributing the attention budget to strengthen image focus within these selected heads.

In summary, we uncover the underlying properties of irrelevant visual tokens and demonstrate that, much like sink tokens in language models, they are unnecessary for the model's functioning. To address this, we propose VAR, which reallocates attention from sink tokens to enhance focus on the image. Experimental results show that VAR improves the overall performance of LMMs across a range of tasks, including general vision-language tasks, visual hallucination tasks, and vision-centric tasks. Notably, VAR can be applied to various models without additional training, models,

or inference steps. This suggests that the existing LMMs can readily benefit from our approach to further enhance their multimodal capabilities by intensifying their attention to images. Our work presents an effective method to address the issue of insufficient image attention and offers a new perspective on understanding the attention mechanisms within LMMs.

## 2 RELATED WORK

**Visual attention in large multimodal models.** In large multimodal models (LMMs), the attention mechanism between text and images plays a pivotal role in incorporating visual information into the text responses. As such, the model's focus on images is typically represented as a visual attention map (Aflalo et al., 2022; Stan et al., 2024). However, recent findings suggest that LMMs exhibit certain unintuitive behaviors in their visual attention patterns. Specifically, LMMs tend to disproportionately focus on a few visual tokens (Woo et al., 2024; Arif et al., 2024), with some tokens receiving high attention weights regardless of the corresponding text token (An et al., 2024). Additionally, recent works have shown that LMMs often fail to adequately attend to visual information overall (Chen et al., 2024; Liu et al., 2024d). To address this issue, visual contrastive decoding (Leng et al., 2024; Favero et al., 2024) has been proposed, which contrasts the outputs of two models—one with and one without visual input to encourage greater reliance on visual cues. Further, other approaches (Zhang et al., 2024b; Zhu et al., 2024) enhance this by increasing the attention weights assigned to images, ensuring that visual information receives sufficient focus.

**Attention sink in language models.** Attention sink is an intriguing phenomenon in language models, where certain *sink tokens* with limited semantic meaning (e.g., 'BOS', '.', ',', '\n', etc.) receive disproportionately high attention weights (Xiao et al., 2023; Ferrando & Voita, 2024). Background tokens, which contain little information, in vision transformers also exhibit similar behavior (Darcet et al., 2023), suggesting that attention sink is a common phenomenon across different modalities. Although sink tokens receive substantial attention weights, they contribute minimally to the model's overall predictions (Kobayashi et al., 2020; Bondarenko et al., 2023). Recent research suggests that attention sink arises from the massive activation of specific dimensions within the hidden states of sink tokens, which occurs prior to the high attention allocation (Sun et al., 2024a; Cancedda, 2024). Gu et al. (2024) further investigated the factors that contribute to the emergence of attention sink. Additionally, Yu et al. (2024) recalibrated the attention weights assigned to sink tokens in specific attention heads to elicit more accurate responses from language models. We extend the concept of attention sink to the multimodal domain by introducing the idea of a *visual attention sink* in LMMs.

## 3 PRELIMINARIES

LMMs typically consist of a visual encoder, a projector, and a large language model. Visual encoder and projector extract visual features from images and project them into text-aligned representations. As shown in the left side of Fig. 2, the large language model receives three types of input: (1) system instructions, (2) visual features from the image, and (3) text including the user's query and preceding context. Then, the model generates responses in an autoregressive manner. In this paper, we refer to the discrete inputs to the large language model, as well as the embeddings within it, as *tokens*.

Let the indices of system tokens, visual tokens, and text tokens be denoted as $\mathcal{I}_{\mathsf{sys}}, \mathcal{I}_{\mathsf{vis}}, \mathcal{I}_{\mathsf{txt}}$, respectively, which are subsets of the indices of all input tokens $\mathcal{I}$. The input is processed through $L$ transformer blocks, each of which consists of multi-head attention (MHA) and feed-forward network (FFN):

$$\hat{\boldsymbol{x}}_i^\ell = \sum_{h=1}^{H} \mathrm{MHA}^{\ell,h}(\boldsymbol{x}_i^{\ell-1}) + \boldsymbol{x}_i^{\ell-1}, \quad \boldsymbol{x}_i^\ell = \mathrm{FFN}^\ell(\hat{\boldsymbol{x}}_i^\ell) + \hat{\boldsymbol{x}}_i^\ell, \tag{1}$$

where $\boldsymbol{x}_i^{\ell-1} \in \mathbb{R}^D$ is the input of the $i$-th token in the $\ell$-th layer and $\hat{\boldsymbol{x}}_i^\ell, \boldsymbol{x}_i^\ell \in \mathbb{R}^D$ are the output of MHA and FFN, respectively.

We now focus on MHA, which enables interactions between different tokens. Following Elhage et al. (2021), the individual input $\boldsymbol{x}_i^{\ell-1}$ interact with the previous tokens $\boldsymbol{X}_{\leq i}^{\ell-1} = \{\boldsymbol{x}_0^{\ell-1}; \cdots; \boldsymbol{x}_i^{\ell-1}\}$ as

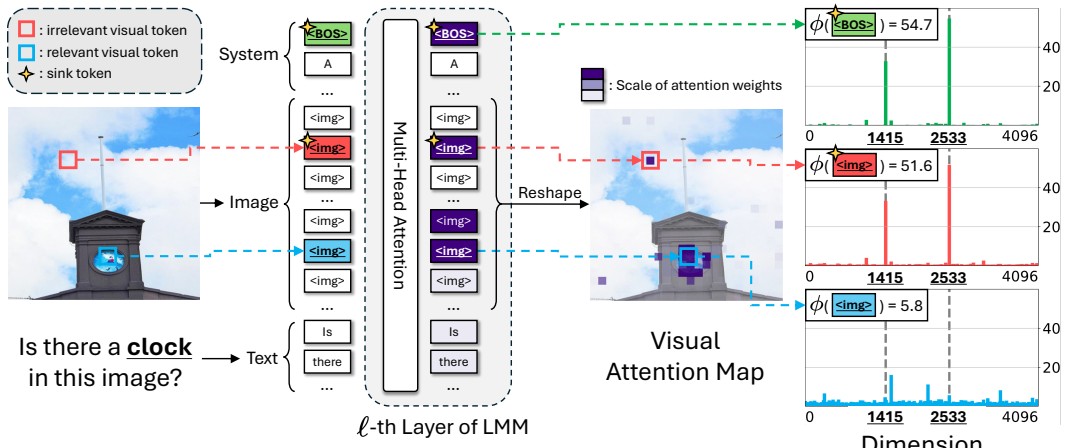

Figure 2: **Illustration of typical architecture of LMMs and investigation of visual attention sink.** A large multimodal model receives the image and text as inputs. Each text token interacts with the visual tokens through the attention mechanism in the transformer decoder. We can visualize the interaction in the form of an attention map. We discover that irrelevant visual tokens (marked as red boxes) in the attention map have massive activation in specific dimensions of hidden states, while relevant visual tokens (marked as blue boxes) do not. Well-known sink tokens (e.g., 'BOS') in language models also exhibit identical patterns in the hidden states.

follows:

$$\text{MHA}^{\ell,h}(\boldsymbol{x}_i^{\ell-1}) = \sum_{j \leq i} \alpha_{i,j}^{\ell,h} \boldsymbol{x}_j^{\ell-1} \boldsymbol{W}_{OV}^{\ell,h}, \quad \boldsymbol{\alpha}_i^{\ell,h} = \text{softmax}\left(\frac{(\boldsymbol{x}_i^{\ell-1}\boldsymbol{W}_Q^{\ell,h})(\boldsymbol{X}_{\leq i}^{\ell-1}\boldsymbol{W}_K^{\ell,h})^\top}{\sqrt{D_k}}\right) \quad (2)$$

where $\boldsymbol{W}_{OV}^{\ell,h} \in \mathbb{R}^{D \times D}$ is the output projection matrix, and $\boldsymbol{W}_Q^{\ell,h}, \boldsymbol{W}_K^{\ell,h} \in \mathbb{R}^{D \times D_k}$ are the query and key projection matrices, respectively. $\boldsymbol{\alpha}_i^{\ell,h}$ are the attention weights from $\boldsymbol{X}_{\leq i}^{\ell-1}$ to $\boldsymbol{x}_i^{\ell-1}$ ($\sum_{j \leq i} \alpha_{i,j}^{\ell,h} = 1$). Eq. 2 indicates that the attention weight $\alpha_{i,j}^{\ell,h}$ can be interpreted as a measure of the extent to which the LMM attends to $\boldsymbol{x}_j^{\ell-1}$ while processing $\boldsymbol{x}_i^{\ell-1}$. As we study how text tokens interact with visual tokens to generate responses, we focus on the attention weights from visual to text tokens, i.e., $\alpha_{i,j}^{\ell,h}$ ($i \in \mathcal{I}_{\text{txt}}, j \in \mathcal{I}_{\text{vis}}$) and investigate them in the form of visual attention map.

## 4 VISUAL ATTENTION SINK

In LMMs, to generate responses that consider visual information, text tokens "see" the image through the attention mechanism in the transformer decoder. Attention from visual tokens (key) to a text token (query) is interpreted as the individual text token's focus on the visual information. Based on this interpretation, we can investigate attention weights from visual tokens to a text token in the form of a visual attention map. Visual attention maps can express the interaction between text and visual tokens in LMMs. Fig. 1 shows the visual attention map between specified text tokens and visual tokens. The model is expected to focus only on the visual tokens related to the text token.

However, the model also attends to some visual tokens which are irrelevant to the corresponding text token, as reported in previous studies (Woo et al., 2024; An et al., 2024). For example, as shown in the top-right of Fig. 1, the model assigns high attention weights to the visual tokens (red boxes) unrelated to the text token banana. Also, the irrelevant visual tokens exist in fixed locations, regardless of the specific text token. This consistent pattern suggests that the irrelevant visual tokens have their own inherent properties causing their appearance. We are interested in the property behind the appearance of these irrelevant visual tokens and understanding their meaning in LMMs.

In the following sections, we find that the irrelevant visual tokens in the visual attention map stem from the massive activation of specific dimensions of hidden states. This phenomenon is analogous to the *attention sink* in language models (Xiao et al., 2023; Sun et al., 2024a), where the model assigns large attention weights to tokens with limited semantic meaning (e.g., BOS). We refer to this phenomenon as *visual attention sink* and further analyse its characteristics.

## 4.1 Investigating the Property of Irrelevant Visual Tokens

We divide the visual tokens with high attention weights in the visual attention map into two categories: *irrelevant visual tokens* and *relevant visual tokens*. Irrelevant visual tokens are visual tokens that are not related to the corresponding text token. In contrast, relevant visual tokens are the visual tokens that are related to the corresponding text token. Fig. 2 illustrates the example of irrelevant and relevant visual tokens as red and blue boxes, respectively.

**How to distinguish irrelevant visual tokens?** We focus on that the irrelevant visual tokens emerge consistently in fixed locations, regardless of the text token. As shown in the bottom-left of Fig. 1, whether the text token is `knife` or `cup`, the model consistently attends to the same irrelevant visual tokens. This observation suggests that irrelevant visual tokens appear not due to the text tokens but as a result of their own inherent properties. Therefore, we examine the hidden states of the irrelevant tokens to investigate their unique property. The right side of Fig. 2. shows the hidden states of the irrelevant visual token (red) and the relevant visual token (blue), as well as the 'BOS' token (green).

**Irrelevant visual tokens have high activation in specific dimensions.** We observe that the hidden states of the irrelevant visual tokens exhibit massive activation in specific dimensions while the relevant visual tokens do not. Also, the dimensions that are highly activated in the irrelevant visual tokens are identical to those of the 'BOS' token, which is known as the representative sink token in language models (Sun et al., 2024a). This observation indicates that the irrelevant visual tokens are closely related to the attention sink.

To extend and formalize this observation further, we check the massive activation values of specific dimensions, called *sink dimensions* $\mathcal{D}_{\mathsf{sink}}$, in the hidden states of the tokens. $\mathcal{D}_{\mathsf{sink}}$ is a set of fixed dimensions that are determined by the base language model of LMMs. For instance, LLaMA-2 (Touvron et al., 2023), the base language model for LLaVA-1.5-7B (Liu et al., 2024a), has $\mathcal{D}_{\mathsf{sink}} = \{1415, 2533\}$. We validate that the sink dimensions in LMMs are consistent with those in base language models in Appendix A.1 and utilize the sink dimensions reported in Sun et al. (2024a). Given a hidden state $\boldsymbol{x} \in \mathbb{R}^D$ of a token, we denote the sink dimension value as follows:

$\phi(\boldsymbol{x}) = \max_{\breve{d} \in \mathcal{D}_{\mathsf{sink}}} \left| \boldsymbol{x}[\breve{d}] / \sqrt{\frac{1}{D} \sum_{d=1}^{D} \boldsymbol{x}[d]^2} \right|$, where $\boldsymbol{x}[d]$ is the $d$-th dimension of the hidden state.

The hidden states are normalized by the root mean square of the dimensions for stability and we only consider the maximum value among the sink dimensions. As shown in the rightmost side of Fig. 2, the sink dimension value of the irrelevant visual token (red) is significantly higher than that of the relevant visual token (blue).

**Sink dimension value can separate irrelevant visual tokens from relevant visual tokens.** We incorporate sink dimension value to discriminate the irrelevant visual tokens from the relevant visual tokens. For visual tokens, we plot the pairwise value of sink dimension value and the corresponding attention weights in Fig. 3(a). Detailed experimental settings are described in Appendix D.3. The sink dimension value distribution of visual tokens with high attention weights is clearly separated into two groups: one with a low sink dimension value and the other with a high sink dimension value. From this analysis, we now define the visual tokens with high sink dimension value as *visual sink tokens*, noting that they are closely related to the attention sink in language models.

Specifically, we set a threshold $\tau$ to divide the distribution in Fig. 3(a) and define the indices of the sink tokens as $\breve{\mathcal{I}}^\ell = \{ j \in \mathcal{I} \mid \phi(\boldsymbol{x}_j^{\ell-1}) \geq \tau \}$, where $\boldsymbol{x}_j^{\ell-1}$ is the input hidden state of the $j$-th token in the $\ell$-th layer. In the subsequent analysis, we set $\tau = 20$. We note that the definition of $\breve{\mathcal{I}}^\ell$ also encompasses all indices of sink tokens, including both visual and text tokens. We denote the visual sink tokens as $\breve{\mathcal{I}}_{\mathsf{vis}}^\ell = \breve{\mathcal{I}}^\ell \cap \mathcal{I}_{\mathsf{vis}}$, where $\mathcal{I}_{\mathsf{vis}}$ is the set of indices of visual tokens. We refer to other visual tokens as *visual non-sink tokens* and denote them as $\mathcal{I}_{\mathsf{vis}} \setminus \breve{\mathcal{I}}_{\mathsf{vis}}^\ell$ for convenience. While the definition of visual sink tokens $\breve{\mathcal{I}}_{\mathsf{vis}}^\ell$ also includes the tokens with low attention weights as shown in Fig. 3(a), they minimally contribute the model due to their low attention weights. Thus, we can neglect them in the subsequent analysis.

## 4.2 Analysing the Characteristics of Visual Sink Tokens

As a next step, we analyse the characteristics of the visual sink tokens. Specifically, we conduct the experiments to validate that the visual sink tokens have analogous characteristics to the sink tokens in

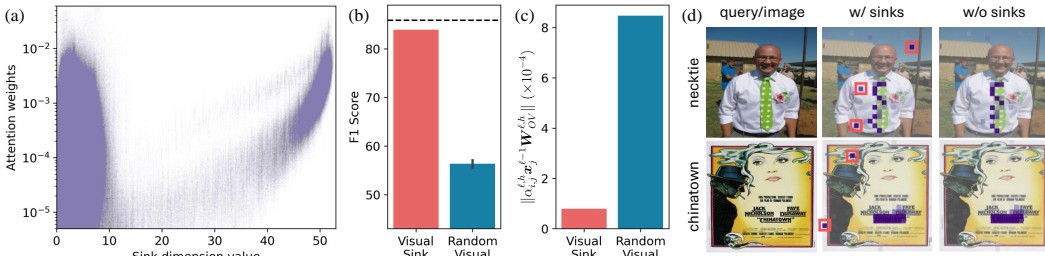

Figure 3: **Analysis of visual sink tokens.** (a) Scatter plot of sink dimension values and attention weights of visual tokens. (b) Performance comparison between masking visual sink tokens and masking the same number of random visual tokens. Dashed line indicates the performance of the original model. (c) Average attention contributions of visual sink tokens and random visual tokens. (d) Visual attention maps with and without visual sink tokens, where the visual sink tokens are highlighted in red boxes.

language models. The sink token itself does not substantially affect the model's response (Kobayashi et al., 2020; Bondarenko et al., 2023; Yu et al., 2024; Gu et al., 2024). We validate whether the visual sink tokens also do not contribute to the model's output by (1) evaluating the performance of the model with the visual sink tokens masked and (2) measuring the mechanistic contribution of the visual sink tokens to the residual stream[1].

**Token Masking Experiment.** To evaluate the impact of the visual sink tokens on the model's output, we mask the attention from the visual sink tokens to the text tokens. This manipulation makes the model not receive any information from the visual sink tokens. As shown in Fig. 3(b), masking the visual sink tokens has little impact on the model's performance. In contrast, masking the same number of random visual tokens leads to a significant performance drop. This result demonstrates that visual sink tokens contribute negligibly to the model's responses.

**Contribution Analysis.** We further investigate the mechanistic contribution of the visual sink tokens to the residual stream. Concretely, we measure attention contributions (Kobayashi et al., 2020; Basu et al., 2024) from the visual sink tokens to the residual stream of the text tokens, which are computed as $\|\alpha_{i,j}^{\ell,h} \boldsymbol{x}_j^{\ell-1} \boldsymbol{W}_{OV}^{\ell,h}\|$, where $i \in \mathcal{I}_{\text{txt}}, j \in \mathcal{I}_{\text{vis}}$ (see Eq. 2 for derivation). Fig. 3(c) shows that the visual sink tokens exhibit significantly lower attention contributions to the residual stream than the other visual tokens. We also qualitatively confirm that the irrelevant visual tokens are clearly filtered out by the definition of visual sink tokens, as illustrated in Fig. 3(d).

**Further discussion on visual attention sink.** In order to explore visual attention sinks more thoroughly, we conduct further analysis on the visual sink tokens and discuss their characteristics in Appendix A.2. Here, we summarize the key takeaways. (1) Visual sink tokens are mostly located in the background, which is less informative. This observation is similar to the findings in ViT (Darcet et al., 2023). Moreover, considering that attention sink in language models occurs in less semantically meaningful tokens (e.g., ',', '\n') (Ferrando & Voita, 2024; Yu et al., 2024), the visual sink tokens also resemble the findings in language models. (2) We discover that visual sink tokens exhibit massive activation at the same dimension $\mathcal{D}_{\text{sink}}$ as text sink tokens. This evidence suggests that the formation of visual sink tokens and text sink tokens shares the same underlying mechanism inherited from the base language models. In summary, some visual tokens are less semantically meaningful, and LMMs treat them as visual sink tokens, similar to the behavior of language models. We leave the investigation of how visual tokens are recognized as sink tokens during training as future work.

## 4.3 SURPLUS ATTENTIONS IN VISUAL ATTENTION SINK: CAN WE RECYCLE THEM?

Our experiment reveals that the visual sink tokens do not contribute to the model's output, even though they have high attention weights. This motivates us to consider the attention weights to the sink tokens as free resources that can be recycled as an "attention budget". Recent studies have shown that LMMs tend to insufficiently attend to the image compared to text, which possibly leads to suboptimal performance in vision-language tasks (Chen et al., 2024; Liu et al., 2024d). This problem can be alleviated by compensating the attention to the image from the attention budget.

---

[1]More Details on experimental settings are described in Appendix D.3.

| | Image and Questions | Heads w/ **high** visual non-sink ratio | | Heads w/ **low** visual non-sink ratio | |
|---|---|---|---|---|---|
| Example 1 | Is there a **skateboard** in this image? | | | | |
| Example 2 | Is there a **red coat** in the image? | | | | |

Figure 4: **Visualization of the attention heads sorted by visual non-sink ratio.** We show some attention heads with high visual non-sink ratio (left) and low visual non-sink ratio (right). The attention heads with high visual non-sink ratio tend to focus on the visual tokens relevant to the corresponding text token. On the other hand, the attention heads with low visual non-sink ratio have vague attention patterns. The attention heads with high visual non-sink ratio are selected as the image-centric heads.

Furthermore, visual sink tokens can be utilized to calculate true image content. Although visual sink tokens receive high attention weights, they do not have semantic meaning related to the corresponding text token. Inversely, visual non-sink tokens are closer to the true image content than the visual sink tokens. Hence, we can exploit the attention assigned to the visual non-sink tokens as a measure of how much the attention head focuses on the image. We apply this concept to select the specific attention heads that focus on the image in the subsequent section.

## 5 VISUAL ATTENTION REDISTRIBUTION

In this section, we introduce Visual Attention Redistribution (VAR), a method to intensify the focus on the image of LMMs based on our discussion in Sec. 4.3. Our method consists of two steps: (1) selecting image-centric heads based on visual attention sink (Sec. 5.1) and (2) redistributing the attention budgets from sink tokens to the visual non-sink tokens only in the selected heads (Sec. 5.2). The overview of VAR is illustrated in Fig. 5.

### 5.1 SELECTING IMAGE-CENTRIC HEADS

In Sec. 4, we propose that the insufficient attention weights to the image can be supplemented by redistributing the attention weights from the sink tokens. However, applying redistribution to all attention heads results in a significant performance drop (see Table 4). Given that each attention head in the transformer possibly has a distinct role (Deiseroth et al., 2023; Zhang et al., 2024a; Ge et al., 2024; Zheng et al., 2024), simply redistributing the attention weights in all heads may ignore the role of some attention heads, whose function is not related with the interaction with the image. Therefore, selecting image-centric heads, which are responsible for focusing on the image, should proceed before redistributing the attention weights.

**Visual attention sink can be utilized to select image-centric heads.** Since heads with low attention weights to the visual tokens are evidently not focusing on the image, we only consider the heads with high attention weights to the visual tokens. Specifically, for each layer $\ell$, we initially discard the heads whose sum of attention weights to the visual tokens is less than 0.2. After that, we incorporate visual attention sink to select image-centric heads. If the model allocates high attention weights to the visual sink tokens, the sum of attention weights to the visual tokens can be high even though the head does not focus on the image. Based on the discussion in Sec. 4.3, the proportion of attention weights allocated to the visual non-sink tokens can indicate how much each attention head actually focuses on the important visual information. Therefore, following the notations in Sec. 3, we define visual non-sink ratio $r_i^{\ell,h}$ as:

$$r_i^{\ell,h} = \frac{\sum_{j \in \mathcal{I}_{\text{vis}} \setminus \check{\mathcal{I}}_{\text{vis}}^\ell} \alpha_{i,j}^{\ell,h}}{\sum_{j \in \mathcal{I}_{\text{vis}}} \alpha_{i,j}^{\ell,h}}, \tag{3}$$

where $\mathcal{I}_{\text{vis}}$ and $\mathcal{I}_{\text{vis}} \setminus \check{\mathcal{I}}_{\text{vis}}^\ell$ denotes the set of all visual tokens and the visual non-sink tokens, respectively. When visual non-sink ratio $r_i^{\ell,h}$ is high, we can expect that the attention head $h$ in layer $\ell$ focuses more on the important visual information.

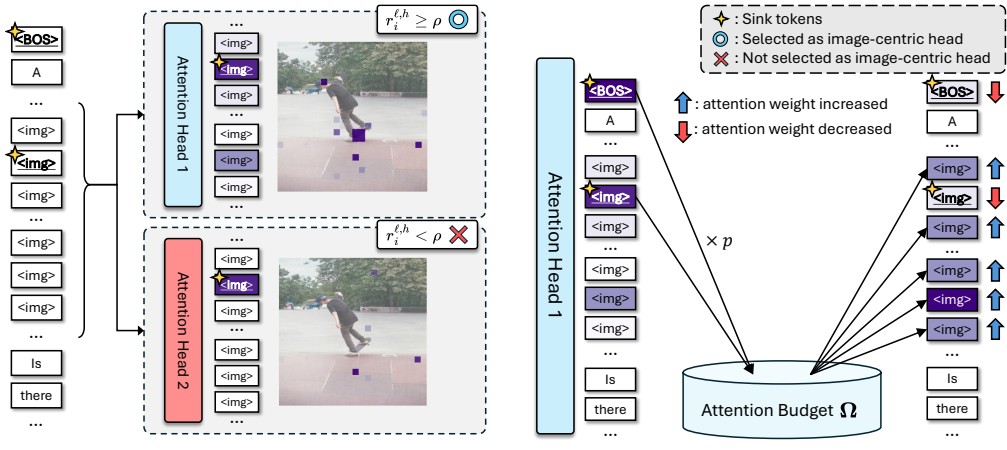

(a) Select image-centric heads  (b) Redistribute attention weights

Figure 5: **Overview of Visual Attention Redistribution (VAR).** (a) We select image-centric heads by evaluating visual non-sink ratio; heads with $r_i^{\ell,h} \geq \rho$ are chosen as image-centric heads. (b) VAR redistributes surplus attention weights from sink tokens to visual non-sink tokens. The attention budget $\mathbf{\Omega}$ accumulates a portion $p$ of attention from sink tokens. Finally, visual non-sink tokens receive attention from $\mathbf{\Omega}$.

**Heads with high visual non-sink ratio focus on the important region.** To validate the effectiveness of visual non-sink ratio $r_i^{\ell,h}$, We sort and visualize the attention heads based on the visual non-sink ratio in Fig. 4. We find that the heads with high visual non-sink ratio are more likely to concentrate on the important visual tokens, which are related to a given text token. On the other hand, the heads with low visual non-sink ratio exhibit a sparse and scattered attention pattern to the various visual tokens. We select attention heads with $r_i^{\ell,h} \geq \rho$ as the *image-centric heads*. Fig. 5(a) illustrates the selection process. Here, $\rho$ is a hyperparameter that controls the number of selected heads. We further investigate the characteristics of the image-centric heads in Appendix A.3.

## 5.2 REDISTRIBUTING ATTENTION WEIGHTS

After selecting the image-centric heads, we redistribute the attention weights from the sink tokens to the visual non-sink tokens in the selected heads. Fig. 5(b) illustrates the redistribution process. We first accumulate the portion $p$ of the attention weights from the sink tokens into the *attention budget* $\mathbf{\Omega}$. The portion $0 \leq p \leq 1$ controls the amount of attention weights to redistribute. For brevity, we will omit the superscript $\ell, h$ hereafter. The attention weights of the sink tokens decrease to $\breve{\alpha}_{i,j} = (1 - p) \cdot \alpha_{i,j}$ and attention budget is calculated as $\mathbf{\Omega} = p \cdot \sum_{j \in \breve{\mathcal{I}}} \alpha_{i,j}$.

Then, we allocate the attention budget to the visual non-sink tokens (i.e., $j \in \mathcal{I}_{\text{vis}} \setminus \breve{\mathcal{I}}_{\text{vis}}$). Inspired by Yu et al. (2024), we redistribute the attention weights considering the relative importance of the visual tokens. After redistributing the attention weights, the attention weights to the visual non-sink tokens are updated as follows:

$$\breve{\alpha}_{i,j} = \alpha_{i,j} + \mathbf{\Omega} \cdot \frac{\alpha_{i,j}}{\sum_{j \in \mathcal{I}_{\text{vis}} \setminus \breve{\mathcal{I}}_{\text{vis}}} \alpha_{i,j}}. \tag{4}$$

This method ensures that the sum of the attention weights remains equal to 1 after redistribution ($\sum_{j \leq i} \breve{\alpha}_{i,j} = 1$), thereby preserving the overall distribution. Note that the redistribution of attention weights is applied to all text tokens $i \in \mathcal{I}_{\text{txt}}$, including the instructions and the generated responses.

## 6 EXPERIMENTS

### 6.1 EXPERIMENTAL SETTINGS

**Model settings.** As our method only modifies the attention of the LMM, VAR can be simply applied to various LMMs without additional training, models, or inference steps. We employ LLaVA-1.5-7B,

Table 1: Benchmark results on general vision-language task.

| Model | VQA$^{v2}$ | GQA | VizWiz | SQA$^I$ | VQA$^T$ | MME | MMB$^{en}$ | SEED$^I$ | LLaVA$^W$ | MM-Vet |
|---|---|---|---|---|---|---|---|---|---|---|
| LLaVA-1.5-7B | 78.5 | 62.0 | 50.0 | 66.8 | 58.2 | 1495.5 | 64.3 | 58.6 | 65.4 | 31.1 |
| + Ours | **78.6** | **63.5** | **53.7** | **67.3** | **58.6** | **1513.8** | **65.1** | **60.7** | **68.1** | **33.7** |
| LLaVA-1.5-13B | 80.0 | 63.3 | 53.6 | 71.6 | 61.3 | 1501.2 | 67.7 | 61.6 | 72.5 | 36.1 |
| + Ours | **81.2** | **64.9** | **57.2** | **72.2** | **62.1** | **1534.3** | **68.1** | **62.3** | **74.1** | **38.4** |
| LLaVA-1.5-HD-13B | 81.8 | 64.7 | 57.5 | 71.0 | 62.5 | 1500.1 | 68.8 | 62.6 | 72.0 | 39.4 |
| + Ours | **82.0** | **65.1** | **58.8** | **71.3** | **63.0** | **1505.2** | **69.5** | **63.3** | **73.8** | **40.6** |
| VILA-13B | 80.8 | 63.3 | 60.6 | 73.7 | 66.6 | 1507.1 | 70.3 | 62.8 | 73.0 | 38.8 |
| + Ours | **81.2** | **63.6** | **64.2** | **74.7** | **67.3** | **1512.7** | **71.7** | **63.0** | **75.7** | **39.7** |
| Qwen2-VL-7B | 82.5 | 64.5 | 65.4 | 74.1 | 84.3 | 1672.3 | 83.0 | 77.9 | 75.6 | 63.2 |
| + Ours | **82.8** | **64.7** | **67.7** | **74.2** | **84.9** | **1688.5** | **83.3** | **78.1** | **77.3** | **63.5** |
| InternVL2-8B | 82.0 | 63.2 | 63.0 | 74.2 | 77.3 | 1648.1 | 81.7 | 76.2 | 73.2 | 60.0 |
| + Ours | **82.5** | **63.5** | **65.1** | **74.7** | **78.0** | **1655.4** | **82.3** | **77.1** | **75.1** | **61.2** |

Table 2: Benchmark results on visual hallucination task.

| Model | CHAIR | | POPE (all) | | MMHal | |
|---|---|---|---|---|---|---|
| | $C_S\downarrow$ | $C_I\downarrow$ | F1↑ | Acc.↑ | Score↑ | Hall.↓ |
| LLaVA-1.5-7B | 45.0 | 14.7 | 85.90 | 84.76 | 2.36 | 51.0 |
| + Ours | **43.2** | **13.8** | **86.53** | **85.87** | **4.26** | **45.1** |
| LLaVA-1.5-13B | 20.6 | 6.2 | 85.90 | 85.47 | 2.42 | 44.3 |
| + Ours | **17.3** | **5.1** | **86.12** | **86.58** | **4.38** | **42.7** |
| LLaVA-1.5-HD-13B | 42.9 | 13.2 | 87.1 | 85.0 | 2.35 | 43.7 |
| + Ours | **40.6** | **12.8** | **87.7** | **87.9** | **4.15** | **43.1** |
| VILA-13B | 31.0 | 8.8 | 84.2 | 83.58 | 2.40 | 44.7 |
| + Ours | **29.7** | **8.0** | **85.1** | **85.4** | **4.19** | **42.9** |
| Qwen2-VL-7B | 30.5 | 8.4 | 87.0 | 87.5 | 2.41 | 44.1 |
| + Ours | **30.1** | **8.2** | **87.4** | **88.2** | **4.09** | **43.5** |
| InternVL2-8B | 32.4 | 9.7 | 86.9 | 87.8 | 2.45 | 43.9 |
| + Ours | **31.8** | **9.3** | **87.5** | **88.6** | **4.17** | **42.7** |

Table 3: Benchmark results on vision-centric task.

| Model | MMVP | CV-Bench$^{2D}$ | CV-Bench$^{3D}$ |
|---|---|---|---|
| LLaVA-1.5-7B | 3.33 | 56.8 | 58.4 |
| + Ours | **9.33** | **57.6** | **59.0** |
| LLaVA-1.5-13B | 24.7 | 58.2 | 58.4 |
| + Ours | **28.0** | **59.6** | **59.9** |
| LLaVA-1.5-HD-13B | 36.0 | 62.7 | 65.7 |
| + Ours | **39.1** | **63.8** | **66.8** |
| VILA-13B | 23.1 | 58.6 | 57.9 |
| + Ours | **28.6** | **59.7** | **59.8** |
| Qwen2-VL-7B | 52.1 | 63.5 | 67.6 |
| + Ours | **55.6** | **63.6** | **67.7** |
| InternVL2-8B | 51.3 | 61.8 | 66.8 |
| + Ours | **56.7** | **62.1** | **67.1** |

LLaVA-1.5-13B, LLaVA-1.5-HD-13B (Liu et al., 2024a), VILA-13B (Lin et al., 2024), Qwen2-VL-7B (Wang et al., 2024), and InternVL2-8B (Team, 2024) as our base models.

**Tasks and benchmarks.** We evaluate our method on a wide range of vision-language benchmarks. The benchmarks are divided into three categories: general vision-language task, visual hallucination task, and vision-centric task. (1) *General vision-language task* assesses comprehensive multimodal capabilities of LMMs. We compare our method with the base models across 10 benchmarks. (2) *Visual hallucination task* evaluates whether the response of the model is consistent with the image content to ensure the trustworthiness and reliability of the model. We use CHAIR (Rohrbach et al., 2018), POPE (Li et al., 2023c), and MMHal-Bench (Sun et al., 2023). (3) *Vision-centric task* evaluates visual understanding capabilities, such as determining the spatial relationship between objects in the image. We use MMVP (Tong et al., 2024b), CV-Bench2D, and CV-Bench3D (Tong et al., 2024a). More details on the tasks and benchmarks are provided in the Appendix D.1.

**Implementation details.** We use the same hyperparameters for all the benchmarks in the same task type. We set $\tau = 20$ and $p = 0.6$ for all experimental settings in our experiments. $\rho$ is set to 0.8 for general vision-language task in Table 1, 0.5 for visual hallucination task in Table 2, and 0.9 for vision-centric task in Table 3. We do not modify the attention heads in the last layer, as the last layer is considered to have a specialized role (Lad et al., 2024; Sun et al., 2024b).

## 6.2 MAIN RESULTS

The experimental results on general vision-language task, visual hallucination task, and vision-centric task are presented in Table 1, 2, and 3, respectively. VAR reliably improves the performance of the base models across all benchmarks. Despite the diverse characteristics and evaluation settings of the benchmarks, VAR demonstrates robust performance without specific hyperparameter tuning for each benchmark. Specifically, Table 2 indicates that VAR effectively mitigates visual hallucination across all benchmarks, and Table 3 demonstrates that complex visual understanding capabilities can be enhanced by editing the attention mechanism of the LMMs. It is worth noting that LLaVA-1.5-7B with VAR outperforms vanilla LLaVA-1.5-13B on GQA, VizWiz, MME, and POPE, suggesting that there is enough margin to improve the performance by simply enhancing the focus on the image without increasing the model size.

Table 4: Ablation study on selecting image-centric heads.

| Setting | POPE | | MME | MM-Vet |
|---|---|---|---|---|
| | F1 | Acc. | Score | Score |
| Baseline | 85.9 | 84.8 | 1495.5 | 31.1 |
| w/o Head selection | 0.0 | 0.0 | 0.0 | 0.0 |
| **w/ Head selection (Ours)** | **86.5** | **85.9** | **1513.8** | **33.7** |

Table 5: Ablation study on redistributing attention weights.

| Setting | POPE | | MME | MM-Vet |
|---|---|---|---|---|
| | F1 | Acc. | Score | Score |
| Baseline | 85.9 | 84.8 | 1495.5 | 31.1 |
| Visual + Text | 86.2 | 85.3 | 1503.9 | 33.5 |
| Text | 70.3 | 70.0 | 1497.3 | 26.7 |
| **Visual (Ours)** | **86.5** | **85.9** | **1513.8** | **33.7** |

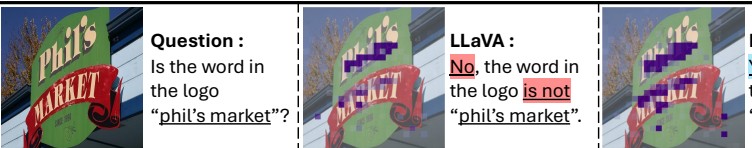

Figure 6: **Qualitative analysis of VAR.** Visual attention maps before and after applying VAR indicate that VAR intensifies the focus on the key visual tokens related to the corresponding text token. As a result, the model can generate more accurate responses by seeing the image more effectively.

### 6.3 ANALYSES AND DISCUSSIONS

**Step-wise ablation studies.** We conduct ablation studies to investigate the effectiveness of each step of VAR on LLaVA-1.5-7B. We validate the necessity of (1) selecting image-centric heads and (2) redistributing attention weights to visual tokens. First, we compare the model's performance when selecting the image-centric heads versus when not selecting them in Table 4. The result shows that redistributing attention weights across all heads may severely impair the functions of some attention heads, leaving the model unable to generate responses (i.e., 0.0 scores w/o head selection in Table 4). As we discussed in Sec. 5.1, head selection is crucial for performance improvement. Second, we compare the model's performance when redistributing the attention budget among both visual and text tokens, only to text tokens, and only to visual tokens (i.e., VAR) in Table 5. Redistributing to text tokens alone yields little improvements, and in some cases, a performance decrease, as the model already sufficiently attends to text tokens. While redistribution to both tokens slightly improves the model's performance, redistribution to visual tokens is the most effective. The result confirms that our method effectively supplements insufficient visual attention by redistributing the attention budget only to the visual tokens.

**Hyperparameters.** We explore the impact of the hyperparameters $\tau$, $\rho$, and $p$ on performance. Below, we outline the key findings, with more detailed results provided in Appendix B.2: (1) VAR is robust to the choice of hyperparameters. Performance consistently improves across all tasks within a reasonable range of hyperparameter values. (2) The optimal values of $\tau$ and $p$ are consistent across all tasks. Therefore, a single value for $\tau$ and $p$ can be applied to all tasks. (3) While the best $\rho$ value varies across tasks, we demonstrate that the optimal $\rho$ remains consistent across different benchmarks within the same task type. Based on this finding, we set a single $\rho$ value for all benchmarks of the same task type.

**Discussion.** We qualitatively analyse the visual attention maps of the base model and VAR in Fig. 6. We observe that VAR effectively redistributes the attention budget to the image, enabling the model to sufficiently focus on the key visual tokens. More qualitative results are provided in Appendix E. Furthermore, our method can be seamlessly incorporated into existing approaches that enhance the performance of LMMs, such as visual contrastive decoding (VCD) (Leng et al., 2024). We provide the experimental results with VCD in Appendix B.1. Overall results provide the evidence that steering the inner attention mechanism is effective way to improve the multimodal capability of LMMs.

## 7 CONCLUSION

This paper discovers the properties and characteristics of visual attention sink in LMMs, demonstrating that the model consistently attends to irrelevant parts of the image. Furthermore, we propose Visual Attention Redistribution (VAR) to emphasize the visual information relevant to the corresponding text tokens by recycling the surplus attention budget from the visual sink tokens. Experimental results imply that LMMs can do better in seeing the image by solely editing the attention map. We hope that our work can contribute to understanding the attention mechanism in LMMs and suggest a new direction for improving the multimodal capabilities of LMMs.

## REPRODUCIBILITY STATEMENT

We ensure reproducibility by providing a detailed explanation of the experimental setup, presenting additional analysis, conducting further experiments on our methods, and offering a comprehensive summary of baseline specifications, all of which are included in Appendix. Our code is included in the supplementary material and all the source codes will be made available to the public.

## ACKNOWLEDGEMENTS

This work was supported in part by the IITP RS-2024-00457882 (AI Research Hub Project), NRF RS-2024-00345806, and NRF RS-2023-00219019.

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

# A  ADDITIONAL ANALYSIS

## A.1  MASSIVE ACTIVATION OF SINK DIMENSIONS

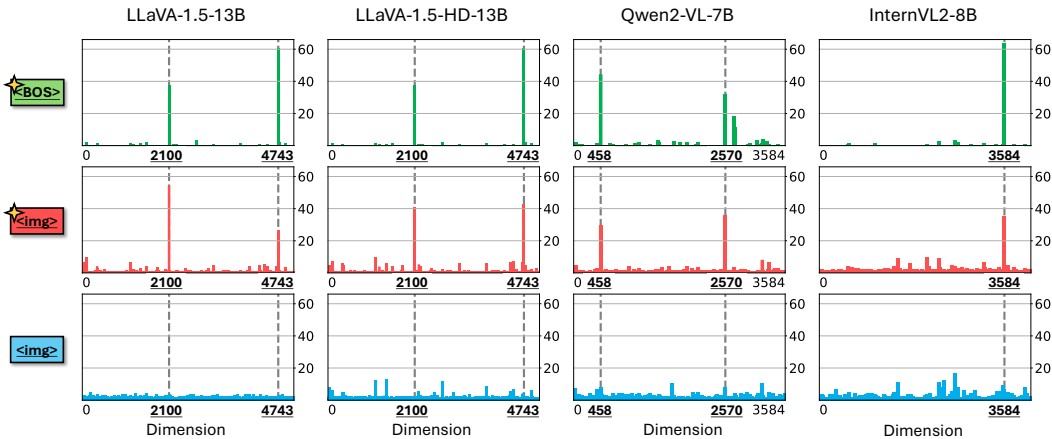

Figure 7: **The hidden states of BOS, visual sink tokens, other visual tokens in various LMMs.** The hidden states of visual sink tokens have massive activation in the same dimensions with the BOS token.

In this section, we demonstrate that sink dimensions $\mathcal{D}_{\text{sink}}$ of visual sink tokens are identical to those of sink tokens in the language models. Based on Sun et al. (2024a), sink tokens have the massive activation in very few fixed dimensions. These dimensions are consistent across different layers and heads and agnostic to the input tokens. Also, the dimensions of massive activation are identical before and after the fine-tuning process. The reason is that the massive activation arises as an innate property during the pre-training process (Darcet et al., 2023; Gu et al., 2024)

We find that the statement is also valid for LMMs. For example, the base LLM of LLaVA-1.5-7B (Liu et al., 2024a) is Vicuna-1.5-7B (Zheng et al., 2023), which has been fine-tuned from LLaMA2-7B (Touvron et al., 2023). Sun et al. (2024a) reports that the dimensions of massive activation in LLaMA2-7B is $\mathcal{D}_{\text{sink}} = \{1415, 2533\}$. Even though the model is fine-tuned with multimodal data, the dimensions of massive activation in LLaVA-1.5-7B are still $\mathcal{D}_{\text{sink}} = \{1415, 2533\}$, as shown in Fig. 2. We also check that this observation is consistent across various LMMs in our experiments. Therefore, we set $\mathcal{D}_{\text{sink}}$ based on the pre-trained large language model of LMMs, following the previous work Sun et al. (2024a). Specifically, we use $\mathcal{D}_{\text{sink}} = \{1415, 2533\}$ for LLaVA-1.5-7B, $\mathcal{D}_{\text{sink}} = \{2100, 4743\}$ for LLaVA-1.5-13B, LLaVA-1.5-HD-13B, and VILA-13B (Lin et al., 2024), $\mathcal{D}_{\text{sink}} = \{458, 2570\}$ for Qwen2-VL-7B (Wang et al., 2024), and $\mathcal{D}_{\text{sink}} = \{3584\}$ for InternVL2-8B (Team, 2024).

Similar to Fig. 2, we observe that the visual sink tokens also have massive activation in the same dimensions with BOS token, as shown in Fig. 7. This observation stands for various LMMs with different architectures, training schemes, and scales. The massive activation in the sink tokens is an innate property of the LMMs, and the visual sink tokens exhibit the same pattern as the sink tokens in the language models. The result suggests that the visual sink tokens share similar characteristics with the sink tokens in the language models.

## A.2  FURTHER ANALYSES AND DISCUSSIONS ON VISUAL ATTENTION SINK

We investigate hidden states of visual sink tokens and analyse the characteristics of visual sink tokens in LMMs in Sec. 4 and Appendix A.1. We discover that the visual sink tokens also exhibit a similar pattern as the sink tokens in the language models. This section provides further analyses and discussions on visual attention sink.

**Location of visual sink tokens.** In Sec. 4, we argue that visual sink tokens are unrelated to the main subject of the image. We validate this argument using large-scale segmentation datasets, such as Pascal-VOC (Everingham et al., 2010) and MS-COCO (Lin et al., 2014). In the segmentation

Table 6: **The proportion of visual sink tokens and random visual tokens located in the background regions.** Visual sink tokens are more likely to be located in the background regions compared to randomly selected visual tokens. The experiment is conducted on LLaVA-1.5-7B.

| Token Type | Pascal-VOC | MS-COCO |
|---|---|---|
| Visual Sink Tokens | 90.5% | 93.7% |
| All Visual Tokens | 82.9% | 90.5% |

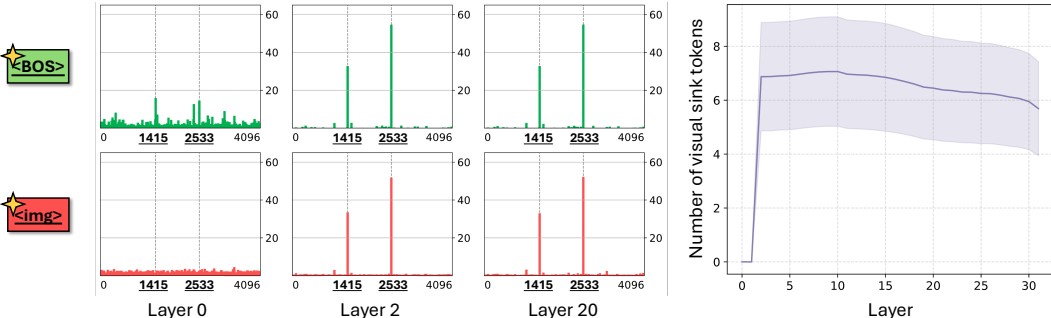

Figure 8: **Layer-wise analysis of visual sink tokens.** (a) Layer-wise hidden states of visual sink tokens. The massive activation of hidden states emerges in the early layers of LMMs. The phenomenon is consistent with the sink tokens in the language models. (b) Number of visual sink tokens in each layer. The visual sink tokens emerge in the early layers and persist until the last layer. The experiment is conducted on LLaVA-1.5-7B and MME Dataset.

data, we treat all regions except the main object as background. After obtaining the visual attention map between object text tokens and visual tokens, we calculate the proportion of visual sink tokens located within the background regions. We compare the portion of visual sink tokens that are not related to the main subject of the image with the portion of all visual tokens that are not related to the main subject of the image, as shown in Table 6. The results indicate that the visual sink tokens are more likely to be located in the background regions than other visual tokens, supporting the conclusion that visual sink tokens are unrelated to the main subject of the image. Furthermore, considering that the background regions are semantically less meaningful, the visual sink tokens are similar to the sink tokens in ViT (Darcet et al., 2023) and language models (Ferrando & Voita, 2024; Xiao et al., 2023; Yu et al., 2024). Specifically, sink tokens in ViT are located in the background regions, and sink tokens in language models have limited semantic meanings (e.g., 'BOS', '.', ',', '\n', etc.).

**Layer-wise analysis.** To investigate when visual sink tokens emerge in the LMMs, we conduct a layer-wise analysis of visual sink tokens. We find that massive activation of hidden states emerges in the early layers of LMMs. For example, as shown in Fig. 8, massive activation emerges in layer 2 of LLaVA-1.5-7B. We also count the number of visual sink tokens in each layer. The visual sink tokens emerge in the early layers and persist until the last layer. This result is consistent with the previous work on sink tokens in the language models (Sun et al., 2024a). Cancedda (2024) shows that feedforward network (FFN) in the early layers is responsible for the attention sink of BOS token in the language models. Early FFNs may write massive activation to the hidden states of sink tokens, and the massive activation can be preserved in the subsequent layers due to the residual connections. Our finding suggests that the massive activation of the visual sink token shares similar causes with the sink token in the language models.

**Further discussions and future works.** Considering the results obtained so far, visual sink tokens are semantically less meaningful and are located in the background regions. These tokens get massive activation from the early layers, resembling the behavior of BOS token. However, the mechanisms behind how LMMs select visual sink tokens and how visual attention sinks emerge during the training process remain open questions. In vision models and language models, sink tokens emerge as a result of enough training on massive training data (Darcet et al., 2023; Gu et al., 2024). Since multimodal models are built upon pre-trained language models, the visual attention sink phenomenon in multimodal models is more likely inherited from the language models. Sup-

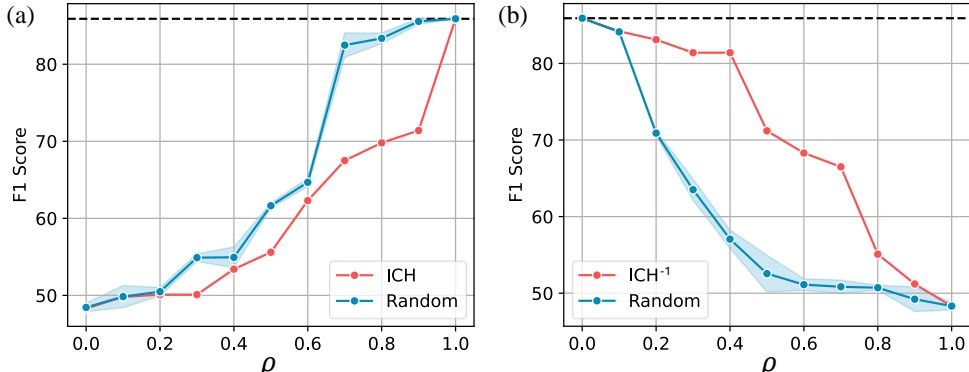

Figure 9: **Ablation study on the importance of image-centric heads (ICHs).** We evaluate the performance of LLaVA-1.5-7B with different $\rho$ on POPE benchmark. (a) The performance drops more significantly when ablating the image-centric heads than when ablating the random heads. (b) On the other hand, the performance remains relatively stable when ablating the complementary heads.

porting evidence for this is that the sink dimensions in multimodal models are identical to those in their base language models, as discussed in Appendix A.1. Therefore, investigating how LMMs identify certain visual tokens as visual sink tokens during the multimodal training process could be an intriguing direction for future research.

### A.3 ANALYSIS ON IMAGE-CENTRIC HEADS

In Sec. 5.1, we select image-centric heads based on $r_i^{\ell,h}$ and utilize them to emphasize the important visual information in LMMs. In this section, we explore the characteristics of image-centric heads.

**Image-centric heads are crucial for visual information processing of LMMs.** We validate the importance of image-centric heads (ICHs) by conducting an ablation study. Specifically, we adjust $\rho$, which is a hyperparameter that controls the number of selected heads. If $\rho$ is high, the threshold for selecting the image-centric heads is high and the number of selected heads is low. To evaluate the contribution of the image-centric heads, we blind the visual information in the image-centric heads. Specifically, we mask the attention from the visual tokens to the text tokens in the image-centric head $(\ell, h)$, using a setting similar to that described in Sec. 4.2. We compare the performance degradation when ablating the image-centric head versus when ablating random heads. We also evaluate the performance degradation when ablating the complementary heads, which are the heads that are not selected as the image-centric heads.

The results are shown in Fig. 9. In Fig. 9(a), the performance drops more significantly when ablating the image-centric heads than when ablating the random heads for each $\rho$. The result indicates that the image-centric heads are crucial for processing visual information in LMMs. In Fig. 9(b), when ablating the complementary heads, the performance remains relatively stable throughout different $\rho$. This result suggests that the other heads are not responsible for processing visual information. The results demonstrate that the image-centric heads selected by visual non-sink ratio $r_i^{\ell,h}$ are indeed essential for integrating visual information into the text responses of LMMs.

**Text tokens related to visual information have more image-centric heads.** Furthermore, we investigate the relationship between the text tokens and the image-centric heads. We note that the number of selected heads are not fixed for all text tokens. We calculate the portion of image-centric heads for each text token. We find that the text tokens related with visual information have more image-centric heads. For example, as shown in Fig. 10, while Sub (the subword token of 'Subway') has more image-centric heads than in. The reason is that the text token Sub requires visual information to generate the text token, while the text token in can be inferred from the question without visual information. This result suggests that the image-centric heads are dynamically selected based on the extent to which the text token requires visual information. Therefore, VAR emphasizes important visual information only when it is required by the text token, considering the context of the text token.

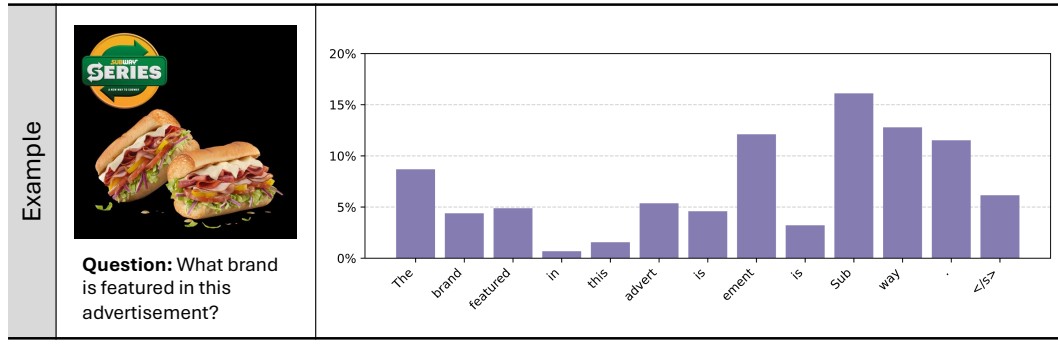

Figure 10: **Relationship between text tokens and image-centric heads.** We calculate the portion of image-centric heads for each text token. Text tokens related with visual information have more image-centric heads. The experiment is conducted on LLaVA-1.5-7B and LLaVA-Bench (In-the-Wild).

## B    MORE EXPERIMENTS ON VISUAL ATTENTION REDISTRIBUTION

### B.1    VISUAL CONTRASTIVE DECODING: A CASE STUDY ON INCORPORATING VISUAL ATTENTION REDISTRIBUTION INTO OTHER METHODS

Table 7: Evaluation results of VCD with VAR.

| Methods | POPE | MME | GQA |
|---|---|---|---|
| Baseline | 85.9 | 1495.5 | 62.0 |
| + VCD | 87.3 | 1580.7 | 66.2 |
| **+ VCD + VAR** | **87.6** | **1597.2** | **67.1** |

In Sec. 6.3, we discuss the potential of applying VAR to other methods which enhance the performance of LMMs in other directions. In this section, we provide a case study on incorporating VAR into the Visual Contrastive Decoding (VCD) (Leng et al., 2024). VCD mitigates object hallucinations in LMMs through contrastive decoding. They require two models: a model with original visual input and a model with distorted visual input. By contrasting the outputs of the two models, VCD effectively emphasizes the visual information and reduces the object hallucinations.

Both our method and VCD aim to enhance the focus on the visual information in LMMs. However, VCD requires two models and additional inference steps, while VAR can be applied to a single model without additional inference steps. Also, we can incorporate VAR into VCD to further enhance the focus on the visual information. Specifically, we contrast the outputs of the model with VAR and the model with distorted visual input. By contrasting the outputs, we expect that two methods can complement each other and further improve the performance of LMMs.

We evaluate the performance of VCD with VAR on the MME (Yin et al., 2023), POPE (Li et al., 2023c), and GQA (Hudson & Manning, 2019) benchmarks. The experiment is conducted on LLaVA-1.5-7B. As shown in Table 7, VCD with VAR improves the performance of VCD on all benchmarks. The results suggest that VAR can be incorporated into various methods to enhance the focus on the visual information in LMMs and further improve the comprehensive performance of LMMs.

### B.2    HYPERPARAMETER ANALYSIS

In this section, we investigate the impact of hyperparameters on the performance of VAR. The hyperparameters include $\tau$, $\rho$, and $p$. First, we randomly sample 10% of the samples from the benchmark and use the partial samples as a "pseudo-validation set" to determine hyperparameters. Second, we apply these hyperparameters to the 100% of the samples in the benchmark and evaluate the impact of the hyperparameters on the performance. The results show that VAR works robustly across reasonable ranges of hyperparameters.

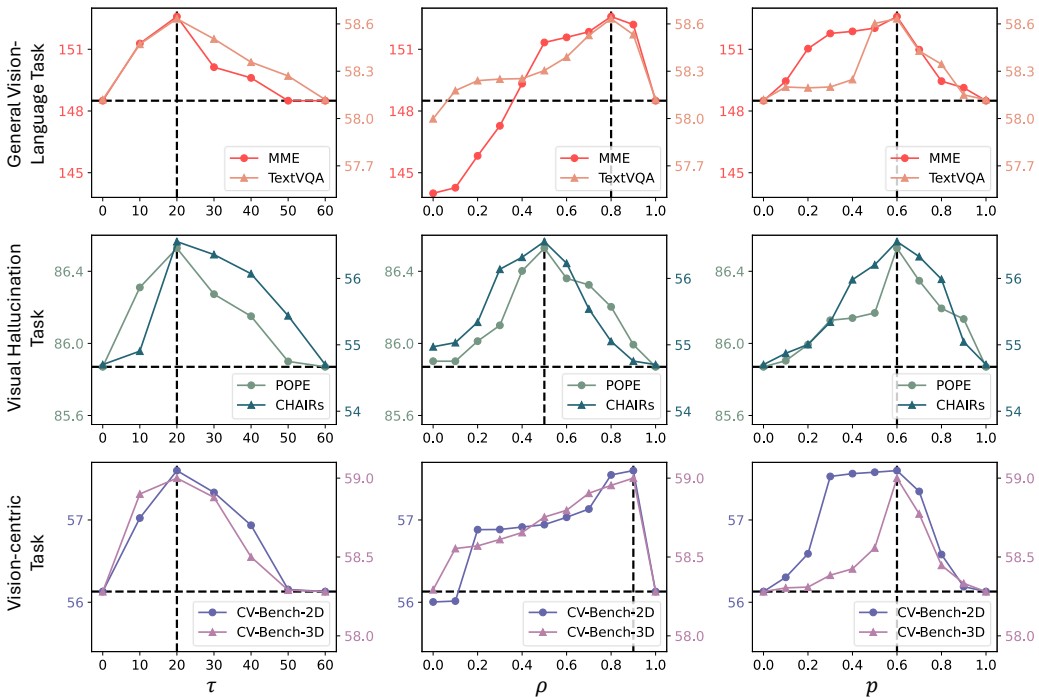

Figure 11: **Determining hyperparameters using 10% of the samples in the benchmark.** We evaluate LLaVA-1.5-7B on the benchmarks with 10% of the samples and determine the hyperparameters. The same optimal hyperparameter values can be obtained from different benchmarks in the same task type.

**Determination of hyperparameters.** Since most benchmarks lack training or validation sets, we alternatively determine hyperparameters using a single benchmark per task and apply these hyperparameters across all benchmarks within the same task type. To mitigate the risk of overfitting to specific benchmarks, we allocate a random 10% of the samples as a pseudo-validation set for hyperparameter tuning. Additionally, we use a different benchmark within the same task type to validate the consistency of the hyperparameters across various benchmarks. We also apply another benchmark in the same task type to determine hyperparameters for validating the consistency of the hyperparameters across different benchmarks. Specifically, we use MME (Yin et al., 2023) and TextVQA (Singh et al., 2019) for the general vision-language task, POPE (Li et al., 2023c) and CHAIR (Rohrbach et al., 2018) for the visual hallucination task, and CV-Bench2D and CV-Bench3D (Tong et al., 2024a) for the vision-centric task. Note that for CHAIR, we report $1 - \text{CHAIR}_S$, as a lower score indicates better performance in this benchmark.

The results are shown in Fig. 11. The results show that the same optimal hyperparameter values can be obtained from different benchmarks in the same task type. Especially, the optimal values of $\tau$ and $p$ are consistent across all tasks, reducing the need for specific hyperparameter settings. Although the optimal $\rho$ value varies across tasks, the optimal $\rho$ remains consistent across different benchmarks within the same task type. The results indicate that the hyperparameters are not overfitted to the specific benchmark and VAR is applicable to various tasks with minimal tuning. Based on the results, we set $\tau = 20$ and $p = 0.6$ for all experimental settings in our experiments. $\rho$ is set to $0.8$ for the general vision-language task, $0.5$ for the visual hallucination task, and $0.9$ for the vision-centric task. We use the same $\rho$ value for all the benchmarks in the same task type.

**Impact of hyperparameters.** We evaluate the impact of hyperparameters on the performance of VAR using MME (Yin et al., 2023), POPE (Li et al., 2023c), and CV-Bench (Tong et al., 2024a) as representative benchmarks for the general vision-language task, visual hallucination task, and vision-centric task, respectively. The results, shown in Fig. 12, are consistent with the results obtained from the pseudo-validation set. Below, we discuss the individual impact of $\tau$, $\rho$, and $p$ on performance in detail.

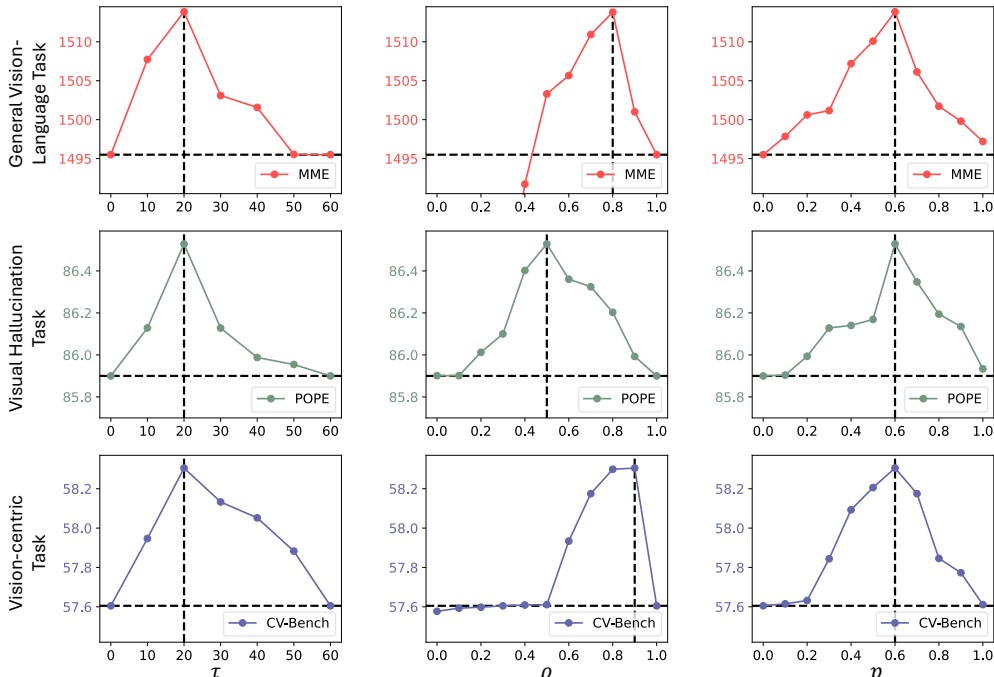

Figure 12: **Impact of hyperparameters on the performance of VAR.** We evaluate LLAVA-1.5-7B on MME (Yin et al., 2023), POPE (Li et al., 2023c), and averaged CV-Bench 2D+3D (Tong et al., 2024a) with different hyperparameters. The results show that VAR is robust to the choice of hyperparameters, and the performance consistently improves the baseline across all tasks within a reasonable range of hyperparameter values.

- **Impact of $\tau$.** Hyperparameter $\tau$ is the threshold for selecting sink tokens. We obtain the best performance when $\tau = 20$. This result is consistent with Sec. 4.1 that the irrelevant visual tokens and relevant visual tokens are clearly separated when $\tau = 20$.

- **Impact of $\rho$.** Hyperparameter $\rho$ is the threshold for selecting image-centric heads. We only find optimal $\rho$ for one benchmark per task. The suitable $\rho$ for each task can vary because the optimal degree of head attention required for each task is different. For example, in the visual hallucination task, the model may need to focus more on the image to reduce hallucination, so the $\rho$ is set to a lower value to select more image-centric heads. In contrast, general vision-language tasks require a balance between text processing and image understanding, so the $\rho$ is set to a higher value to select fewer image-centric heads. However, in the reasonable range of $\rho$ values, the proposed method robustly improves the performance of various LMMs on various benchmarks. For example, if $\rho \geq 0.5$, VAR can consistently improve the performance.

- **Impact of $p$.** Hyperparameter $p$ is the portion of the attention weights from the sink tokens into the attention budget. We find that $p = 0.6$ is the best value for all benchmarks, but different values of $p$ can improve the performance, indicating that VAR is robust to the choice of $p$.

Overall, VAR demonstrates robustness to the selection of hyperparameters. Performance shows consistent improvement across all tasks within a reasonable range of hyperparameter values, indicating that VAR is not highly sensitive to specific hyperparameter choices. This suggests that VAR can enhance the general performance of LMMs even when hyperparameters vary within an plausible range, providing flexibility in its application across different tasks.

## C  LIMITATIONS AND FUTURE WORKS

In this study, we utilize raw visual attention map without post-processing in our method. However, the raw attention map may not be perfect for grounding the object because the model may have seen the object incorrectly (Guan et al., 2024) or the visual encoder may misinterpret the image (Tong et al., 2024b). We can further refine the visual attention map by investigating the localization of the object. Also, although we conduct extensive experiments on various LMMs and benchmarks, it is still worth exploring our methodology on different and larger LMMs (Li et al., 2024; Agrawal et al., 2024). Visual sink tokens can be utilized for various applications. For example, masking visual sink tokens removes noise and produces more reliable and interpretable visual attention map. Moreover, free attention budget from sink tokens can be exploited to emphasize the specific part of the image, depending on the user's interest. In addition, as discussed in Sec. 4.2, understanding how LMMs recognize particular visual tokens as visual sink tokens during multimodal training could be an intriguing direction for future research.

## D  EXPERIMENTAL DETAILS

### D.1  DETAILS OF BENCHMARKS

**General vision-language task.** In Table 1, abbreviations of the benchmarks are used for brevity. We use the following benchmarks for general vision-language task:

- $VQA^{v2}$ (Goyal et al., 2017): Visual question answering v2.0.

- GQA (Hudson & Manning, 2019): Question answering on image scene graphs. We used *test-dev-balanced* set splits for the evaluation.

- VizWiz (Gurari et al., 2018): We used *val* set splits for the evaluation.

- $SQA^I$ (Lu et al., 2022): ScienceQA is a dataset collected from elementary and high school science curricula, consisting of 21,208 multimodal multiple-choice science questions. Out of these, 10,332 questions include image context, 10,220 include text context, and 6,532 include both. Most questions are annotated with lectures (17,795) and detailed explanations (19,184) to provide general knowledge and specific reasoning for the correct answers. The dataset spans three subjects: natural science, language science, and social science, and is organized into 26 topics, 127 categories, and 379 skills.

- $VQA^T$ (Singh et al., 2019): TextVQA. We used *val* set splits for the evaluation.

- MME (Yin et al., 2023): MME serves as a thorough evaluation benchmark for large multimodal models, assessing both their perception and reasoning capabilities. It covers 14 different tasks, including object existence, counting, spatial position, color recognition, poster identification, celebrity recognition, scene understanding, landmark detection, artwork recognition, OCR, commonsense reasoning, numerical calculations, text translation, and code reasoning. We evaluated the model's performance specifically on the perception tasks.

- $MMB^{en}$ (Liu et al., 2023): MMBench-english. MM-Bench consists of around 3,000 multiple-choice questions designed to assess 20 different ability dimensions. Each question has a single correct answer. For evaluation, GPT-4 is used to match the model's prediction with the answer choices, outputting a label (A, B, C, D) as the final prediction. We use the circular evaluation strategy to test if a vision-language model can successfully solve each single problem. In this paper, we use only the English subset for evaluating our method.

- $SEED^I$ (Li et al., 2023a): SEED-Bench-image. SEED-Bench shows that current multimodal large language models (MLLMs) struggle with understanding spatial relationships between objects, even though they perform well on tasks involving overall image comprehension. This benchmark evaluates models on both images and videos using multiple-choice questions, but we focused only on the image-based section for our evaluation.

- $LLaVA^W$ (Liu et al., 2024c): LLaVA-Bench (In-the-Wild) consists of 24 images paired with 60 questions, covering a variety of contexts such as indoor and outdoor environments, memes, paintings, and sketches. The dataset is designed to evaluate how well LMMs handle

more complex tasks and adapt to unfamiliar domains. We perform case studies on this dataset to qualitatively showcase the effectiveness of our proposed VAR method.

- MM-Vet (Yu et al., 2023): MM-Vet assesses a model's ability to engage in visual conversations across various tasks, evaluating both the accuracy and helpfulness of responses using GPT-4. The dataset includes 200 images and 218 questions, each paired with corresponding ground truth answers.

**Visual hallucination task.** We use the following benchmarks in Table 2 to evaluate the visual hallucination performance of the model.

- CHAIR (Rohrbach et al., 2018): We evaluate our algorithm on both captioning and visual question answering (VQA) benchmarks. For captioning, we measure object hallucinations using the CHAIR (Captioning Hallucination Assessment with Image Relevance) metrics. These metrics compare the objects mentioned in the captions with those actually present in the image to assess caption quality. CHAIR has two versions: one calculates the percentage of hallucinated objects in the entire caption ($\text{CHAIR}_I$), while the other measures the percentage of captions that contain at least one hallucinated object ($\text{CHAIR}_S$). $\text{CHAIR}_I$ and $\text{CHAIR}_S$ are expressed by the following equations:

$$\text{CHAIR}_I = \frac{\# \text{ hallucinated objects}}{\# \text{ generated objects}}, \quad \text{CHAIR}_S = \frac{\# \text{ hallucinated captions}}{\# \text{ generated captions}}. \quad (5)$$

- POPE (Li et al., 2023c): Polling-based Object Probing Evaluation (POPE) utilizes data from MSCOCO Lin et al. (2014), A-OKVQA Schwenk et al. (2022), and GQA Hudson & Manning (2019), generating 27,000 query-answer pairs from 500 images per dataset. This benchmark offers a streamlined approach to assess object hallucination by querying large multimodal models (LMMs) on the presence of specific objects in images. Queries are equally divided between existent and non-existent objects (50% each). Negative samples are selected through three strategies: random, popular, and adversarial. In the random setting, missing objects are chosen randomly; in the popular setting, they are picked from a pool of frequently occurring objects; and in the adversarial setting, co-occurring but absent objects are prioritized. Each image is associated with 6 questions, and the evaluation relies on four key metrics: Accuracy, Precision, Recall, and F1 score.
- MMHal-Bench (Sun et al., 2023): MMHal-Bench is an evaluation benchmark designed to assess hallucination in Large Multimodal Models (LMM). It includes 96 challenging questions based on images from OpenImages, along with their corresponding ground-truth answers and image content. Model responses are automatically rated using GPT-4.

**Vision-centric task.** We use the following benchmarks in Table 3 to evaluate the vision-centric performance of the model.

- MMVP (Tong et al., 2024b): MMVP is a dataset of 300 images paired with related questions, curated through a systematic study using the CLIP visual encoder. It consists of image pairs, called CLIP-blind pairs, where the CLIP vision encoder represents two different images with such similar features that it cannot distinguish between them.
- CV-Bench (Tong et al., 2024a): CV-Bench addresses the limitations of existing vision-centric benchmarks by providing 2,638 manually inspected examples. It repurposes standard vision datasets such as ADE20k, COCO, and OMNI3D to evaluate models on traditional vision tasks in a multimodal context. Using the rich ground truth annotations from these benchmarks, natural language questions are formulated to assess the models' fundamental 2D and 3D understanding. CV-Bench measures 2D understanding through tasks like spatial relationships and object counting, and 3D understanding through depth order and relative distance. This dataset is structured in a multiple-choice question answering (MCQA) format, and models are evaluated based on the accuracy of their responses to each question.

## D.2 DETAILS OF EVALUATION SETTING

**Experimental Setting.** All experiments and evaluations are conducted on a single NVIDIA GeForce RTX A6000 48GB GPU. Only the inference step of LMMs is used, without any training.

**GPT-4 evaluation.** LLaVA-Bench (In-the-Wild), MM-Vet, and MMHal-bench are benchmarks that use GPT-4 to evaluate model performance. We conducted evaluations using the gpt-4-0613 version, with a maximal output length set to 1024.

### D.3 DETAILS OF ANALYSIS

In this section, we provide the detailed experimental settings of Sec. 4.

**Investigation of Sink Dimension Values.** In Sec. 4.1, we separate the irrelevant visual tokens from relevant visual tokens based on the sink dimensions, as shown in Fig. 3(a). First, we selected 10 samples from each category, resulting in a total of 100 visual question-answering samples from MME dataset (Yin et al., 2023). We then obtain the attention weights from the visual tokens to the text tokens for each sample. Also, we calculate sink dimension value (see Sec. 4.1) for each visual token. Finally, We then plot the pairwise scatter plot of the sink dimension value and the attention weights for all samples. The scatter plot is shown in Fig. 3(a).

**Token Masking Experiment.** In Sec. 4.2, we conduct token masking experiments to investigate the impact of visual sink tokens on the performance of LMMs. Specifically, we mask the attention from visual sink tokens to text tokens to blind the model to the visual sink tokens. As discussed in Sec. 4.1, the indices of the visual sink tokens are defined as $\check{\mathcal{I}}^{\ell}_{\text{vis}} = \{\, j \in \mathcal{I}_{\text{vis}} \mid \phi(\boldsymbol{x}^{\ell-1}_j) \geq \tau \,\}$, where $\tau$ is a threshold value. We set $\tau = 20$ for experiments.

We employ the procedure introduced in Geva et al. (2023) to mask the attention. Given the query matrix $\boldsymbol{Q} = \boldsymbol{X}^{\ell-1}_{\leq i}\boldsymbol{W}^{\ell,h}_Q \in \mathbb{R}^{N \times D_k}$ And the key matrix $\boldsymbol{K} = \boldsymbol{X}^{\ell-1}_{\leq i}\boldsymbol{W}^{\ell,h}_K \in \mathbb{R}^{N \times D_k}$, the attention weights $\boldsymbol{A}^{\ell,h} \in \mathbb{R}^{N \times N}$ are calculated as follows:

$$\boldsymbol{A}^{\ell,h} = \text{softmax}\left(\frac{\boldsymbol{Q}\boldsymbol{K}^{\top}}{\sqrt{D_k}} + \boldsymbol{M}\right), \tag{6}$$

where $\boldsymbol{M} \in \mathbb{R}^{N \times N}$ is the causal mask. The causal mask is a upper triangular matrix with negative infinity values (i.e., $\boldsymbol{M}_{i,j} = -\infty$ for $i < j$), which prevents the model from attending to the future tokens. The procedure, called *attention knockout*, blocks the attention from $\boldsymbol{x}^{\ell-1}_j$ to $\boldsymbol{x}^{\ell-1}_i$ by setting $\boldsymbol{M}_{i,j} = -\infty$. We set the attention mask values to $-\infty$ from the visual sink tokens to the text tokens as follows:

$$\boldsymbol{M}_{i,j} = -\infty, \quad \forall i \in \mathcal{I}_{\text{txt}}, j \in \check{\mathcal{I}}^{\ell}_{\text{vis}}, \tag{7}$$

where $\mathcal{I}_{\text{txt}}$ is the indices of text tokens and $\check{\mathcal{I}}^{\ell}_{\text{vis}}$ is the indices of visual sink tokens in the layer $\ell$. We then calculate the attention weights $\boldsymbol{A}^{\ell,h}$ and feed them into the subsequent layers.

We also conduct a random-token attention knockout experiment, in which the same number of tokens are masked without considering whether they are sink tokens. For instance, if eight sink tokens are identified at layer $\ell$, we randomly select eight tokens to mask in the corresponding random knockout setting to ensure a fair comparison. Note that randomness is applied differently at each layer. We evaluate the performance of LLaVA-1.5-7B on the POPE dataset (Li et al., 2023c), with results shown in Fig. 3(b).

While attention knockout is widely used to analyze the influence of specific tokens on a model's output (Neo et al., 2024; Kaduri et al., 2024), this experiment has some limitations. First, setting the attention mask values to $-\infty$, which forces the corresponding attention weights to zero, induces out-of-distribution behavior, potentially preventing the token knockout effect from being accurately reflected in the model's output. Second, in random knockout, tokens are selected independently at each layer, leading to inconsistencies across layers compared to sink token knockout. Despite these limitations, we believe the token masking experiments offer valuable insights into the influence of visual sink tokens on the model's performance.

**Contribution Analysis.** In Sec. 4.2, we measure the attribution contributions of visual sink tokens. We compute the average value of $\|\alpha^{\ell,h}_{i,j}\boldsymbol{x}^{\ell-1}_j\boldsymbol{W}^{\ell,h}_{OV}\|$ for all $i \in \mathcal{I}_{\text{txt}}$ and $j \in \check{\mathcal{I}}^{\ell}_{\text{vis}}$. As a baseline, we similarly calculate the average value for all $j \in \mathcal{I}_{\text{vis}}$. Similar to the token masking experiment, we use the LLaVA-1.5-7B model and the POPE dataset (Li et al., 2023c) for the analysis, with results shown in Fig. 3(c).

# E    ADDITIONAL QUALITATIVE RESULTS

Additional qualitative results are provided in Fig. 13, Fig. 14 and Fig. 15.

Although we visualize the attention maps only for a few specified text tokens in Fig. 1, these irrelevant tokens consistently occur in fixed locations across the entire text tokens, including the instructions and the generated responses. To illustrate this, we visualize the attention maps of various text tokens in Fig. 13. The visual attention maps show that the visual sink tokens consistently receive high attention weights from the text tokens, regardless of the content of the text tokens.

Fig. 14 shows the responses generated by the model with and without VAR. We use LLaVA-1.5-7B and LLaVA-Bench (In-the-Wild) (Liu et al., 2024c) benchmark. We can observe that the model with VAR generates more accurate responses than the model without VAR. In particular, the model with VAR considers the visual information more effectively and generates more relevant responses to the image content.

To evaluate the performance of LMMs in open-ended question-answering tasks, we utilize GPT-4 (OpenAI, 2023) to assess the performance of the model in various aspects. We provide the qualitative result of GPT-4 evaluation in Fig. 15.

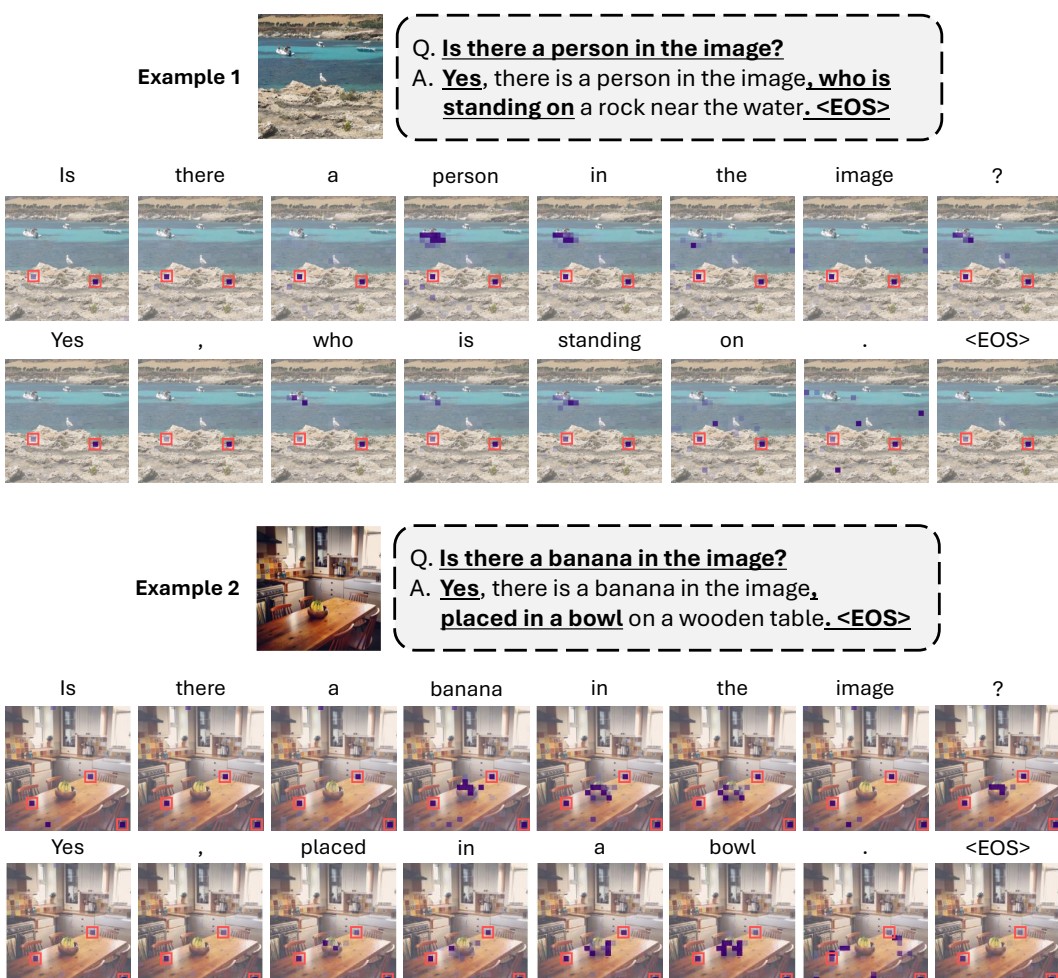

Figure 13: **Visual attention maps of various text tokens in LLaVA-1.5-7B.** The visual sink tokens consistently receive high attention weights from the text tokens, regardless of the content of the text tokens. We underline the text tokens that are visualized, and visual sink tokens are marked with red boxes.

| | | | |
|---|---|---|---|
| **Example 1** | Question: Describe this photo in detail. | LLaVA | The unusual aspect of this image is that a man is standing on a clothesline attached to a yellow car, which is driving down a busy city street. This is not a typical sight, as clotheslines are usually found in residential areas or backyards, and it is not common to see someone standing on a clothesline while a car is in motion. The scene captures an unconventional and unexpected moment, likely drawing attention and curiosity from passersby. |
| | | LLaVA + Ours | The unusual aspect of this image is that a man is standing on a folding table, which is attached to a yellow car, and he is ironing clothes. This is not a typical scene you would expect to see on a busy city street, as ironing clothes is usually done indoors, such as in a home or a laundromat. The presence of the folding table and the man's activity in the middle of the street creates an interesting and unexpected visual. |
| **Example 2** | Question: Show the detailed recipe for this dish. | LLaVA | To prepare this dish, you will need the following ingredients:
1. Shrimp: You can use raw or cooked shrimp, depending on your preference.
2. Lemons: You will need a few whole lemons, sliced into wedges.\n3. Seasonings: You can use salt, pepper, garlic powder, and any other desired seasonings.
Instructions:
1. Preheat your oven to 400! °F (200 °C).
2. Rinse the shrimp and pat dry with a paper towel. *...(omit)* |
| | | LLaVA + Ours | To prepare this dish, you will need fresh lobster tails, lemon wedges, garlic, and parsley. Here's a step-by-step guide:
1. Clean and cook the lobster tails: Remove the claws and knuckles from the lobster tails, and cook them according to your preference. You can boil, steam, or grill the tails until they are cooked through.
2. Prepare the garlic and parsley: Chop the garlic cloves and parsley leaves finely.
3. Prepare the lemon wedges: Cut the lemon wedges into *...(omit)* |
| **Example 3** | Question: This is the logo of LLaVA, Large Language and Vision Assistant, based on the LLaMA architecture. Please explain this logo in detail, and how do you think of its design. | LLaVA | The LLaVA logo features a red and black color scheme, with a large red fire-like element in the center. The fire appears to be burning, giving the logo a dynamic and engaging appearance. The design of the logo is reminiscent of a fire hydrant, which adds a playful and creative touch to the overall design. The combination of the bold red color and the fire-like element creates a striking visual impact, making the logo memorable and easily recognizable. The choice of a fire hydrant-like shape also suggests a connection to the field of artificial intelligence, *...(omit)* |
| | | LLaVA + Ours | The LLaVA logo features a red fire-breathing animal, likely a dragon or a similar creature, with a pair of glasses on its face. The animal is standing on a gray surface, which could be a table or a floor. The design of the logo is quite unique and eye-catching, as it combines the concept of fire and a humorous, anthropomorphic character. The glasses on the animal's face add a touch of whimsy and personality to the logo, making it memorable and appealing. Overall, the LLaVA logo effectively conveys the idea of a vision-based AI assistant while incorporating a fun and engaging visual element. |
| **Example 4** | Question: Can you give a short introduction to this person? | LLaVA | The image features a man with curly hair, a mustache, and a beard, who is wearing a suit and tie. He is smiling and appears to be a well-dressed individual. The man is also wearing glasses, which adds to his distinctive appearance. It is not possible to determine the exact identity of the person from the image alone, but he could be a businessman, a public figure, or someone attending a formal event. |
| | | LLaVA + Ours | The image features a man with curly hair and a distinctive appearance, who is likely a famous scientist or intellectual. He is smiling and appears to be enjoying the conversation with the person next to him. The man's hair is wild and untamed, and he is wearing a suit, which suggests that he is well-dressed for the occasion. The man's presence and attire indicate that he is a prominent figure in the field of science or academia. |

Figure 14: **Qualitative results of the model with and without VAR.** We compare the responses generated by LLaVA-1.5-7B with and without VAR. Our method makes the model incorporate visual information more effectively and generate more accurate responses, which are closely related to the image content. While vanilla LLaVA-1.5-7B incorrectly describes the image ( red highlights ), LLaVA-1.5-7B with VAR generates more accurate responses, which are closely related to the image content ( blue highlights ).

```
input:"""
```
[Context]
A top view of a highway at night. The highway is divided into four sections. From left to right, there are three lanes and four lanes of traffic approaching the camera, and four lanes and three lanes of traffic moving away from the camera. Most of the cars in the four-lane section moving away from the camera have their brake lights on. There are many cars in both the four-lane sections. The traffic is light on the two three-lane sections. The four-lane highway is elevated compared to the three-lane highway. The lights alongside the highway are illuminated. On the right side of the highway, there are trees.

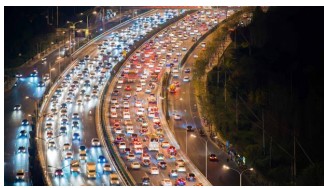

[Question]
What is the problem this city might be facing? What are some possible solutions?

[Assistant 1]
The city might be facing traffic congestion issues, particularly in the four-lane sections of the highway since most cars have their brake lights on, indicating slow or stopped traffic. Possible solutions could include:

1. Implementing traffic management strategies, such as ramp meters to control the rate at which cars enter the highway or optimizing traffic signal timings to improve traffic flow.
2. Encouraging the use of public transportation, carpooling or ride-sharing to reduce the number of cars on the road.
3. Expanding or adding more lanes to ease traffic congestion in the four-lane sections of the highway.
4. Exploring alternative routes to divert traffic from the congested areas.
5. Improving walking and biking infrastructure in the city, thus promoting more sustainable and non-motorized modes of transport.
[End of Assistant 1]

[Assistant 2]
The city is facing a significant traffic problem, as evidenced by the heavy traffic on the highway at night. The image shows a large number of cars and trucks on the highway, with some cars stopped in traffic. This congestion can lead to increased travel times, air pollution, and a higher risk of accidents.\n\nSome possible solutions to address this problem include:

1. Implementing public transportation options, such as buses, trains, or subways, to encourage people to use these services instead of personal vehicles.
2. Encouraging carpooling or ridesharing to reduce the number of cars on the road.
3. Implementing congestion pricing or high-occupancy vehicle (HOV) lanes to discourage single-occupancy vehicles and promote the use of more efficient transportation options.
4. Developing and promoting alternative transportation modes, such as cycling or walking, to reduce the reliance on personal vehicles.
5. Improving traffic management systems, such as implementing smart traffic signals or real-time traffic updates, to help drivers make better decisions and avoid unnecessary delays.

By implementing these solutions, the city can work towards reducing traffic congestion and improving overall transportation efficiency.
[End of Assistant 2]

[System]
We would like to request your feedback on the performance of two AI assistants in response to the user question displayed above. The user asks the question on observing an image. For your reference, the visual content in the image is represented with a few sentences describing the image.
Please rate the helpfulness, relevance, accuracy, level of details of their responses. Each assistant receives an overall score on a scale of 1 to 10, where a higher score indicates better overall performance.
Please first output a single line containing only two values indicating the scores for Assistant 1 and 2, respectively. The two scores are separated by a space.

In the subsequent line, please provide a comprehensive explanation of your evaluation, avoiding any potential bias and ensuring that the order in which the responses were presented does not affect your judgment.
"""

```
GPT4 Output: """
```
9 9

Both Assistant 1 and Assistant 2 provided comprehensive, relevant, and accurate answers to the question. They both identified the potential problem of traffic congestion and suggested similar solutions such as implementing traffic management strategies, encouraging the use of public transportation, carpooling or ride-sharing, and improving walking and biking infrastructure. They also both suggested expanding the highway or finding alternative routes. The level of detail in both responses was high, with each assistant providing multiple potential solutions and explaining how these solutions could alleviate the problem. Therefore, they both receive a score of 9."""

Figure 15: **Qualitative GPT-4 evaluation results.** The example of evaluation of GPT-4 in open-ended generation tasks is illustrated. "Assistant 1" and "Assistant 2" refer to "GPT-4" and "LLaVA-1.5-7B + VAR", respectively.

