# OpenReview forum: "See What You Are Told: Visual Attention Sink in Large Multimodal Models"
_ICLR.cc/2025/Conference — ICLR 2025 Poster_

### Official Review · Reviewer_b7X3 · 2024-10-24

**Soundness:** 3
**Presentation:** 3
**Contribution:** 2
**Rating:** 6
**Confidence:** 4

**Summary:**

In this paper, the authors introduce the phenomenon of visual attention sinks and, through visual analysis, propose that visual attention sink tokens are typically unrelated to the main visual subject information. Furthermore, they demonstrate through experiments how to automatically identify visual sink tokens. Based on this finding, the authors propose a method that utilizes attention budgets to redistribute attention weights. Extensive experiments have shown that this method optimizes both the general visual capabilities of models and the issue of hallucination.

The paper makes two main contributions:
1. It introduces the phenomenon of visual attention sinks and analyzes the behavior of LVLMs.

2. The proposed method enhances the performance of LVLMs with low cost.

**Strengths:**

1. The writing in this paper is very clear, smoothly transitioning from the introduction of the visual attention sink phenomenon to its analysis and the subsequent methodological approach.

2. The concept of visual attention sink proposed in this paper is indeed a direction worthy of deeper exploration within the realm of LVLMs.

3. The experiments in this paper cover a comprehensive range of content and consistently achieve stable improvements across various tasks.

4. The method proposed in this paper is an optimization based on the attention mechanism, making it generalizable and applicable to most LVLMs for experimentation.

**Weaknesses:**

1. The phenomenon of visual attention sink appears to be similar to the summary tokens proposed in OPERA[1]. Summary tokens imply that higher attention weights are assigned to tokens without semantic information, such as ',' and '\n'. The visual sink token seems to suggest that within visual tokens, there are also some tokens that play a similar role to summary tokens.

2. The analysis of the visual attention sink phenomenon lacks experimental validation. For more specific explanations, please refer to Question 1.

3. Although the author's evaluation experiments cover a comprehensive set of benchmarks, there is a lack of evaluation across more LVLMs and a detailed analysis of the experimental results. For more specific explanations, please refer to Question 2.

[1] OPERA: Alleviating Hallucination in Multi-Modal Large Language Models via Over-Trust Penalty and Retrospection-Allocation

**Questions:**

1. Regarding the questions of visual attention sink:

   1.a. I have previously experimented with removing the <bos> token, and the attention sink phenomenon still persisted in the first token. Therefore, I am curious whether new tokens will become visual sink tokens after redistributing the attention weights.

   1.b. The author has visualized tokens with high attention weights in the text, and indeed, for these cases, the visual sink tokens are unrelated to the main subject of the image. However, I believe it is still necessary to prove the trend of visual sink tokens being unrelated to the main subject of the image on a large-scale dataset based on segmentation data.

   1.c. In Table 4.a, the selection of random tokens does not seem to be related to $\tau$, so I want to confirm whether two masked methods with the same $\tau$ value mean that the same number of tokens are masked.

2. Regarding the questions of experimental results:

   2.a. The author only chose LLaVA and VILA for evaluation, but the training data for LLaVA is much smaller compared to high-performance open-source models. Therefore, it is necessary to add one or two such models for evaluation, such as Qwen2-VL and InternVL2.

   2.b. LLaVA-HD has several times more image tokens compared to LLaVA, so I speculate that the author's model should perform better on this backbone. However, from Table 1, it appears that the improvement of the proposed method on LLaVA-HD is not as significant as on LLaVA.

If the author can address these concerns, I will consider raising the score. Additionally, I believe that further analysis on how to utilize this phenomenon during model training could enhance the impact of the article.

---

> ### Author Response · Authors · 2024-11-23
> **Response to Reviewer b7X3 (1/3)**
>
> We thank the reviewer for their constructive feedback and helpful observations. Our detailed responses to the comments are outlined below. If there are any remaining concerns, please inform us, and we will address them promptly.
>
> # W1. Comparison of Visual Attention Sinks and Summary Tokens in OPERA
>
> > W1. The phenomenon of visual attention sink appears to be similar to the summary tokens proposed in OPERA[1]. Summary tokens imply that higher attention weights are assigned to tokens without semantic information, such as ',' and '\n'. The visual sink token seems to suggest that within visual tokens, there are also some tokens that play a similar role to summary tokens.
>
> Thank you for the insightful and interesting comment. The visual sink tokens and the summary tokens may seem similar in terms of having high attention weights, but they are inherently different in nature. The summary tokens, or the anchor tokens in OPERA [1] was originally suggested by [2]. According to [2], the anchor tokens (1) gather the information of demonstrations to form semantic representations for deeper layers in shallow layers and (2) the information from label words is extracted to form the final prediction in deep layers. Therefore, high attention weights to the summary tokens emerge in the middle to later layers. In contrast, the visual sink tokens are observed from the early layers to the late layers. We also tried to investigate the summary tokens (e.g., ‘,’ and ‘.’) on some samples of the CHAIR dataset in LLaVA-1.5-7B. However, there is no massive activation on sink dimensions $\mathcal{D}_{\textsf{sink}} = \lbrace 1415, 2533 \rbrace$, indicating that the summary tokens and visual sink tokens are different. Furthermore, visual sink tokens are almost useless, as discussed in Sec 4.2 and Sec 4.3, while the summary tokens have a crucial role to transfer the information of the whole sentence.
>
> [1] Huang, Qidong, et al. "OPERA: Alleviating Hallucination in Multi-Modal Large Language Models via Over-Trust Penalty and Retrospection-Allocation." Proceedings of the IEEE/CVF Conference on Computer Vision and Pattern Recognition. 2024.
>
> [2] Wang, Lean, et al. "Label Words are Anchors: An Information Flow Perspective for Understanding In-Context Learning." Proceedings of the 2023 Conference on Empirical Methods in Natural Language Processing. 2023.

---

> ### Author Response · Authors · 2024-11-23
> **Response to Reviewer b7X3 (2/3)**
>
> # W2 / Q1. Lack of Experimental Validation for Visual Attention Sink
>
> > 1.a. I have previously experimented with removing the <bos> token, and the attention sink phenomenon still persisted in the first token. Therefore, I am curious whether new tokens will become visual sink tokens after redistributing the attention weights.
>
> Thank you for the insightful and interesting question. Due to the autoregressive nature of LLMs, the former token does not depend on the latter token. Since the visual tokens appear before the text tokens and VAR only redistributes the attention weights while processing the text tokens, the visual sink tokens are preserved after the attention weights are redistributed. Also, new visual sink tokens do not emerge after redistributing the attention weights.
>
> On the other hand, we agree that removing the visual sink tokens and checking whether new visual sink tokens emerge is an interesting direction. Therefore, we have conducted an experiment to remove the visual sink tokens and check whether new visual sink tokens emerge, as the reviewer has experimented with <bos> token. We find that new visual sink tokens do not emerge after removing the visual sink tokens. In fact, except for the first position, the sink tokens in LLMs (e.g., ',' and '\n') also do not persist in the same position when we remove them. Therefore, our observation is coherent with the results in LLMs.
>
> > 1.b. The author has visualized tokens with high attention weights in the text, and indeed, for these cases, the visual sink tokens are unrelated to the main subject of the image. However, I believe it is still necessary to prove the trend of visual sink tokens being unrelated to the main subject of the image on a large-scale dataset based on segmentation data.
>
> We agree that the validation on the segmentation dataset can show stronger evidence for the trend of visual sink tokens being unrelated to the main subject of the image. We have conducted an additional experiment to validate whether the visual sink tokens are unrelated to the main subject of the image. We use Pascal-VOC and MS-COCO datasets, which are widely used large-scale datasets for object segmentation. We compare the portion of visual sink tokens that are not related to the main subject of the image with the portion of all visual tokens that are not related to the main subject of the image. The results are shown below.
>
> | Token Type | Pascal-VOC | MS-COCO |
> | :--------: | :--------: | :-----: |
> | Visual Sink Tokens | 90.5% | 93.7% |
> | All Visual Tokens | 82.9% | 90.5% |
>
> The results indicate that visual sink tokens are more likely to be located in the background regions compared to randomly selected visual tokens, supporting the conclusion that visual sink tokens are unrelated to the main subject of the image. We have added these results to the revised manuscript (Sec 4 and Appendix A.2).
>
> > 1.c. In Table 4.a, the selection of random tokens does not seem to be related to $\tau$, so I want to confirm whether two masked methods with the same $\tau$ value mean that the same number of tokens are masked.
>
> Maybe the reviewer mentioned Figure 4(a) (revised as Figure 3(b)). Yes, the number of tokens to be masked is the same for both methods for fair comparison. We have clarified this in the caption of Figure 3.

---

> > ### Author Response · Authors · 2024-11-23
> > **Response to Reviewer b7X3 (3/3)**
> >
> > # W3 / Q2. More Experiments and Analyses with various LMMs
> >
> > > 2.a. The author only chose LLaVA and VILA for evaluation, but the training data for LLaVA is much smaller compared to high-performance open-source models. Therefore, it is necessary to add one or two such models for evaluation, such as Qwen2-VL and InternVL2.
> >
> > We agree that more evaluation on various models, especially high-performance open-source models, is essential. We have added Qwen2-VL and InternVL2 to the main experiments in Table 1, 2, and 3. The results demonstrate that the proposed method consistently improves the performance on various models.
> >
> > > 2.b. LLaVA-HD has several times more image tokens compared to LLaVA, so I speculate that the author's model should perform better on this backbone. However, from Table 1, it appears that the improvement of the proposed method on LLaVA-HD is not as significant as on LLaVA.
> >
> > The number of visual tokens in LLaVA-HD is larger than that in LLaVA, therefore there are more visual tokens to redistribute attention budget $\boldsymbol{\Omega}$. For evidence, we have compared the ratio of the number of visual tokens to the number of sink tokens in LLaVA and LLaVA-HD. The ratio represents the average number of visual tokens that need to have attention weights allocated per sink token.
> >
> > |      Model       |     Ratio      |
> > | :--------------: | :------------: |
> > |  LLaVA-1.5-13B   | 122.6 (± 27.3) |
> > | LLaVA-1.5-HD-13B | 258.3 (± 60.0) |
> >
> > The results show that the ratio in LLaVA-HD is larger than that in LLaVA, which implies that allocated attention budget per visual token is relatively low in LLaVA-HD. Therefore, the improvement of the proposed method on LLaVA-HD is not as significant as on LLaVA.
> >
> > However, we can allocate more attention weights from sink tokens to visual tokens by adjusting the hyperparameter $p$. Hyperparameter $p$ controls the portion of the attention weights from sink tokens into the attention budget. To show that the performance on LLaVA-HD can be further improved by allocating more attention weights from sink tokens, we have conducted an additional experiment by adjusting the hyperparameter $p$. If there is a margin to improve the performance on LLaVA-HD by allocating more attention weights from sink tokens, the optimal $p$ value would be larger than the default value $p=0.6$. The results are shown below.
> >
> > |             p             | 0 (Baseline) |   0.1   |   0.2   |   0.3   |   0.4   |   0.5   |   *0.6*   |   0.7   |   **0.8**   |   0.9   |    1    |
> > | :-----------------------: | :----------: | :-----: | :-----: | :-----: | :-----: | :-----: | :-------: | :-----: | :---------: | :-----: | :-----: |
> > |     MME (General VQA)     |   1500.10    | 1501.28 | 1501.28 | 1501.28 | 1505.20 | 1505.20 | *1505.20* | 1513.37 | **1513.37** | 1510.84 | 1505.20 |
> > | POPE 10% (Hallucination)  |    87.10     |  87.10  |  87.11  |  87.35  |  87.50  |  87.55  |  *87.70*  |  87.71  |  **87.72**  |  87.50  |  87.43  |
> > | CV-Bench (Vision-Centric) |    64.20     |  64.58  |  64.60  |  65.05  |  65.14  |  65.25  |  *65.30*  |  66.33  |  **66.50**  |  66.28  |  65.35  |
> >
> > The results indicate that the optimal $p$ value is $0.8$, which is larger than the default value $p=0.6$. This result supports our claim that the relatively low improvement on LLaVA-HD is due to the large number of visual tokens and can be further improved by adjusting the hyperparameter $p$. Note that we still report the results with the default $p$ value $p=0.6$ in the main paper for consistency.
> >
> > # + Visual Attention Sink on Training
> >
> > > Additionally, I believe that further analysis on how to utilize this phenomenon during model training could enhance the impact of the article.
> >
> > We agree that further analysis on how to utilize this phenomenon during model training is an interesting direction for research. If we can remove the visual sink tokens during training, the model can focus more on the important tokens. Introducing register tokens or attention bias terms [2] during training can be a potential solution. However, since the base LLM already has the attention sink phenomenon, it is not enough to retrain the LVLM with off-the-shelf LLMs. Rather, we may need to train the LLM from scratch with the register tokens, which is computationally expensive for our current resources. We have added this direction as future work in Sec 7.
> >
> > [1] Darcet, Timothée, et al. "Vision Transformers Need Registers." The Twelfth International Conference on Learning Representations. 2024.
> >
> > [2] Sun, Mingjie, et al. "Massive Activations in Large Language Models." Conference on Language Modeling (COLM), 2024.

---

> > > ### Comment · Reviewer_b7X3 · 2024-11-23
> > >
> > > Thank you for the detailed response, I have decided to raise my score.

---

### Official Review · Reviewer_KaLi · 2024-10-26

**Soundness:** 4
**Presentation:** 4
**Contribution:** 2
**Rating:** 6
**Confidence:** 5

**Summary:**

This paper studies the phenomenon of visual sink tokens in large multi-modal models. That is, the model allocates high attention scores to visual tokens irrelevant to the corresponding text, and these visual tokens consistently appear in fixed locations. This paper finds that visual sink tokens have high activations in specific dimensions, which provides a way to distinguish them from normal tokens. Furthermore, the paper proposes to recycle the attention to meaningful tokens, helping to improve model performances on a wide range of tasks.

**Strengths:**

1. The presentation is really good, first analyzing the property and effects of visual sink tokens, then providing a solution based on these investigations. The figures are also nice to explain relevant concepts and methods.
2. While the phenomenon of sink tokens has been discussed in language models[1] and vision models[2], this paper further extend it to multi-modal models.
3. The proposed method is simple yet effective. The model can be modified without further training.

[1] Xiao, Guangxuan, Yuandong Tian, Beidi Chen, Song Han, and Mike Lewis. "Efficient Streaming Language Models with Attention Sinks." In The Twelfth International Conference on Learning Representations.

[2] Darcet, Timothée, Maxime Oquab, Julien Mairal, and Piotr Bojanowski. "Vision Transformers Need Registers." In The Twelfth International Conference on Learning Representations.

**Weaknesses:**

1. Insufficient references. There is a work [1] on sink tokens in vision models that is not cited or discussed.
2. Insufficient discussion on sink tokens. Sink tokens have been observed in different kinds of models, including language models, vision models, and multi-modal models. What's the relationship between sink tokens in different kinds of models? Are they just similar phenomenons, or are there special properties of multimodal models? Moreover, previous works claim that sink tokens exist because of excessive attention scores, so why does it work to recycle these attention scores? From the experiment perspective, the paper is also encouraged to apply previous methods to multi-modal models.
3. Heavy hyper-parameter tuning for specific tasks. It seems that different tasks are highly sensitive to the selection of $\rho$. Then is there a train/val/test dataset split? If not, are these hyper-parameters tuned according to the test set? Will these hyper-parameters overfit the test set?

[1] Darcet, Timothée, Maxime Oquab, Julien Mairal, and Piotr Bojanowski. "Vision Transformers Need Registers." In The Twelfth International Conference on Learning Representations.

**Questions:**

1. As mentioned in the weakness, the paper should discuss the relationship between sink tokens in language models and vision models, from the perspective of token properties, emerging mechanisms and proposed methods.
2. How does the method determine which layer to apply VAR? The supplementary shows that there are less visual sink tokens in the first layer. However, the experiments say that VAR is not applied to the last layer. That does not sound reasonable.

---

> ### Author Response · Authors · 2024-11-23
> **Response to Reviewer KaLi (1/2)**
>
> We appreciate the reviewer's thoughtful feedback and valuable suggestions. We provide a detailed response to the reviewer's comments below.
>
> # W1 / W2 / Q1. Insufficient Discussion on Sink Tokens
>
> > W1. Insufficient references. There is a work [1] on sink tokens in vision models that is not cited or discussed.
>
> > W2. Insufficient discussion on sink tokens. Sink tokens have been observed in different kinds of models, including language models, vision models, and multi-modal models. What's the relationship between sink tokens in different kinds of models? Are they just similar phenomenons, or are there special properties of multimodal models? Moreover, previous works claim that sink tokens exist because of excessive attention scores, so why does it work to recycle these attention scores? From the experiment perspective, the paper is also encouraged to apply previous methods to multi-modal models.
>
> > Q1. As mentioned in the weakness, the paper should discuss the relationship between sink tokens in language models and vision models, from the perspective of token properties, emerging mechanisms and proposed methods.
>
> We appreciate the reviewer for pointing out the insufficient discussion on the relationship between sink tokens in different kinds of models. Sink tokens in vision models, language models, and multimodal models are similar phenomenons, but those in multimodal models are slightly different.
>
> (**Token Properties**) High attention weights and massive activation of sink tokens are common properties in different models. We also find that visual sink tokens are more likely to be located in the background region (unrelated to the main objects). This characteristic is consistent with the findings in vision models [1] and language models [3]. Specifically, sink tokens in ViT are located in the background regions, and sink tokens in language models have limited semantic meanings. Therefore, we conclude that the tokens with less semantic meanings are more likely to be sink tokens in various models.
>
> (**Emerging Mechanisms**) In vision models and language models, sink tokens emerge as a result of massive training data [1, 2]. Since multimodal models are based on the pre-trained language models, visual attention sink phenomenon in multimodal models are more likely to be inherited from language models. The evidence is that sink dimensions in multimodal models are identical to those in base language models, as discussed in Appendix A.1.
>
> (**Proposed Methods**) As the reviewer mentioned, sink tokens exist because of excessive attention scores. However, the excessive attention scores are useless for the model's prediction, as we discussed in Sec 4.2 and 4.3. Recycling these attention scores is effective and safe because it compensates for the low attention weights on visual tokens and does not modify the original attention distribution except for the sink tokens. We tried to apply previous methods such as [3] to multimodal models, but there was no improvement in the performance. The Visual + Text setting in Table 5 can be considered as a soft version of [3] but it is not as effective as the proposed method (VAR).
>
> Overall, we agree that the paper should discuss the relationship between sink tokens in different kinds of models more thoroughly. We have added [1] to the related work and discussed the relationship between sink tokens in different models in Sec 4.2 and Appendix A.2. If there are any remaining concerns, please inform us, and we will address them promptly.
>
> [1] Darcet, Timothée, et al. "Vision Transformers Need Registers." The Twelfth International Conference on Learning Representations. 2024.
>
> [2] Gu, Xiangming, et al. "When Attention Sink Emerges in Language Models: An Empirical View." 2024.
>
> [3] Yu, Zhongzhi, et al. “Unveiling and Harnessing Hidden Attention Sinks: Enhancing Large Language Models without Training through Attention Calibration." Forty-first International Conference on Machine Learning. 2024.

---

> > ### Author Response · Authors · 2024-11-23
> > **Response to Reviewer KaLi (2/2)**
> >
> > # W3. Heavy Hyper-Parameter Tuning of $\rho$
> >
> > > W3-1. Heavy hyper-parameter tuning for specific tasks. It seems that different tasks are highly sensitive to the selection of $\rho$.
> >
> > We appreciate the reviewer for raising this question. We acknowledge that the optimal value of $\rho$ is task-dependent. However, reasonable $\rho$ values (e.g., $\rho \geq 0.5$) consistently improve the performance of the proposed method on various tasks. Therefore, we argue that the proposed method is still applicable to various tasks in plausible ranges of $\rho$ values. We have discussed the hyperparameter $\rho$ more thoroughly in Sec 6.3. and Appendix B.2.
> >
> > > W3-2. Then is there a train/val/test dataset split? If not, are these hyper-parameters tuned according to the test set? Will these hyper-parameters overfit the test set?
> >
> > Most recent LVLM benchmarks do not include dataset splits because they primarily focus on evaluating a model's inference performance. Therefore, we use the test set (which is the benchmark itself) to determine hyperparameters. As the reviewer mentioned, tuning hyper-parameters on the test set can lead to overfitting. To mitigate this concern, we have further validated that the same optimal hyperparameter values can be obtained from partial samples of the benchmark (i.e., only 10% of the samples) or another benchmark in the same task type. The results indicate that the hyperparameters are not overfitted to the specific benchmark. We have included the results of the experiments with 10% of the samples in the benchmark in Appendix B.2.
> >
> > # Q2. Determination of The Layer to Apply VAR
> >
> > > Q2. How does the method determine which layer to apply VAR? The supplementary shows that there are less visual sink tokens in the first layer. However, the experiments say that VAR is not applied to the last layer. That does not sound reasonable.
> >
> > We apologize for the confusion. Early layers and last layer are known to have special roles in the model, rather than processing interaction between related tokens [1]. Early layers aggregate information in an n-gram-like manner [2, 3], while the last layer refines the token prediction [1, 4]. Based on this prior knowledge, we applied VAR to the layers where it is expected to be effective. In the early layers, the number of sink tokens is already limited and VAR has little impact. Therefore, we do not need to manually restrict VAR to the early layers. In contrast, we manually disable the application of VAR in the last layer.
> >
> > [1] Lad, Vedang, Wes Gurnee, and Max Tegmark. "The Remarkable Robustness of LLMs: Stages of Inference?." 2024.
> >
> > [2] Ferrando, Javier, and Elena Voita. “Information Flow Routes: Automatically Interpreting Language Models at Scale.” Proceedings of the 2024 Conference on Empirical Methods in Natural Language Processing. 2024.
> >
> > [3] Gurnee, Wes, et al. "Finding Neurons in a Haystack: Case Studies with Sparse Probing." Transactions on Machine Learning Research. 2024.
> >
> > [4] Sharma, Pratyusha, Jordan T. Ash, and Dipendra Misra. "The Truth is in There: Improving Reasoning in Language Models with Layer-Selective Rank Reduction." The Twelfth International Conference on Learning Representations. 2024.

---

> ### Comment · Reviewer_KaLi · 2024-11-25
>
> Thanks for the authors' response. I have also read reviews of other reviewers. Most of my concerns have been addressed. I'm already on the positive side, so I will maintain my original score.

---

### Official Review · Reviewer_P74f · 2024-10-30

**Soundness:** 2
**Presentation:** 2
**Contribution:** 3
**Rating:** 3
**Confidence:** 4

**Summary:**

The paper explores the phenomenon of visual attention sinks in large multimodal models (LMMs). It identifies that these models often allocate high attention weights to irrelevant visual tokens during text-image processing, similar to "attention sinks" observed in language models. This issue results in diminished attention to crucial visual information, which can hinder the model’s multimodal understanding. To address this, the authors propose a method called Visual Attention Redistribution (VAR), which reallocates attention from irrelevant "sink" tokens to relevant visual tokens. The VAR approach first identifies image-centric attention heads that naturally focus on visual content, then redistributes the excess attention from sink tokens to strengthen attention on the actual image content.

**Strengths:**

- Insightful Identification of Visual Attention Sink: The authors make an important discovery by identifying the phenomenon of visual attention sinks in LMMs, showing that these models often allocate high attention weights to irrelevant visual tokens. This observation highlights a key inefficiency in multimodal attention mechanisms and contributes to a deeper understanding of model behaviors.

- Effective Attention Redistribution Technique: The proposed Visual Attention Redistribution (VAR) method is both innovative and practical. By reallocating attention from sink tokens to relevant visual tokens, VAR improves focus on key image content, enhancing multimodal model performance without additional training or inference overhead, making it easily applicable across different LMMs.

- Broad Experimental Validation Across Tasks: The paper rigorously validates VAR across a range of multimodal tasks, including visual question answering, visual hallucination reduction, and spatial understanding tasks. This broad evaluation demonstrates the robustness and adaptability of the method, showcasing its effectiveness in various vision-language scenarios.

**Weaknesses:**

- The results in Figure 4(a) are essential for validating the effectiveness of the sink token filtering strategy, yet they are challenging to interpret. If the randomly selected tokens are drawn from all input tokens, comparing them with visual sink tokens becomes less fair, making it difficult to substantiate the filtering's effectiveness. Conversely, if the random tokens are sampled exclusively from visual tokens, the experimental results appear unreasonable. Even without visual input, the model’s accuracy on the POPE benchmark should remain above 50%, yet the figure shows a final result of 0. Furthermore, when $\tau < 10$, the tokens being pruned are predominantly standard visual tokens. Despite this, a significant difference in F1-Score persists between the two pruning strategies, contradicting the results seen when $\tau > 10$. Without a clear explanation of the experimental findings in Figure 4(a), the credibility of the sink token filtering strategy is undermined.

- The scope of the attention redistribution strategy remains ambiguous. Specifically, which tokens are targeted by this strategy? In practice, attention redistribution could apply to both input instruction tokens and output tokens, but the implementation should align with the analytical findings. If the strategy is applied only to specific tokens, it is crucial to specify how these tokens are selected; if applied to all output tokens, the analysis should demonstrate stable consistency in the attention assigned to sink tokens. However, this consistency is not currently evident in the analysis.

**Questions:**

1. In Figure 1, does the attention map show the attention scores between visual tokens and a specified text token, or between visual tokens and the output token?

2. In the experiment shown in Figure 4(a) on the POPE benchmark, are the "random tokens" randomly selected from only the visual tokens? If not, and they are instead selected from all tokens, wouldn’t this comparison be unfair? If the tokens are indeed chosen only from visual tokens, how is it possible that the model's predictions drop to zero when masking all visual tokens? On the POPE dataset, LLaVA can maintain over 50% accuracy even without image input, so this experimental result is difficult to understand.

3. Regarding the selection of image-centric attention heads, is the VAR strategy applied to all text tokens, or only to specific text tokens, or only to output tokens? If the experiment aligns with the analysis that VAR strategy only applied to specific object token, like "clock" in "Is there a clock in this image?", how are these particular tokens identified? If applied to all text tokens, is the visual token sink phenomenon consistent across all text tokens, which seems unlikely?

4. During inference, is the attention redistribution performed layer by layer? Specifically, does the forward pass first obtain original results, identify image-centric attention heads, then update attention weights, and proceed with attention weight redistribution in the next layer?

If the author can address my second and third questions, I will raise my rating.

---

> ### Author Response · Authors · 2024-11-23
> **Response to Reviewer P74f (1/3)**
>
> Thank you for your detailed and thoughtful review. First, we apologize for the lack of clarity in the experimental setup details, descriptions, and incomplete results that may have made the reading uncomfortable. The review was very helpful for improving our paper, making many parts more clarified or concrete. We provide a detailed response to the reviewer's comments below. If there are any remaining concerns, please inform us, and we will address them  promptly.
>
> # Q1. Detailed Clarification of Figure 1
>
> > Q1. In Figure 1, does the attention map show the attention scores between visual tokens and a specified text token, or between visual tokens and the output token?
>
> The attention map in Fig 1 shows the attention scores between visual tokens and a specified text token. We have clarified this in the caption of the revised manuscript.
>
> # W1 / Q2. Concerns Regarding Interpretation of Figure 4(a) Results
>
> > W1. The results in Figure 4(a) are essential for validating the effectiveness of the sink token filtering strategy, yet they are challenging to interpret. If the randomly selected tokens are drawn from all input tokens, comparing them with visual sink tokens becomes less fair, making it difficult to substantiate the filtering's effectiveness. Conversely, if the random tokens are sampled exclusively from visual tokens, the experimental results appear unreasonable. Even without visual input, the model’s accuracy on the POPE benchmark should remain above 50%, yet the figure shows a final result of 0. Furthermore, when $\tau < 10$, the tokens being pruned are predominantly standard visual tokens. Despite this, a significant difference in F1-Score persists between the two pruning strategies, contradicting the results seen when $\tau > 10$. Without a clear explanation of the experimental findings in Figure 4(a), the credibility of the sink token filtering strategy is undermined.
>
> > Q2. In the experiment shown in Figure 4(a) on the POPE benchmark, are the "random tokens" randomly selected from only the visual tokens? If not, and they are instead selected from all tokens, wouldn’t this comparison be unfair? If the tokens are indeed chosen only from visual tokens, how is it possible that the model's predictions drop to zero when masking all visual tokens? On the POPE dataset, LLaVA can maintain over 50% accuracy even without image input, so this experimental result is difficult to understand.
>
> Thank you for raising these questions. We will address them separately in three parts. Note that Figure 4(a) is changed to Figure 3(b) in the revised manuscript.
>
> ## A. Random Token Selection: Visual Tokens or All Tokens?
>
> The "random tokens" in the experiment shown in Fig 4(a) are randomly selected from only the visual tokens. As the reviewer pointed out, we agree that masking all input tokens including text tokens is unfair. We have clarified that in the figure, caption and main text of the revised manuscript.
>
> ## B. F1 Score of 0 in Figure 4(a) (revised as Fig 3(b))
>
> We acknowledge that an F1 Score of 0 seems contradictory with common sense. The reason for this is the attention masking method we used. There are two mainstream methods for attention masking: before softmax (set the attention mask to -inf) [1, 2, 3] and after softmax (set the attention weight to 0) [4, 5, 6]. Both are widely used in literature. We chose the latter method for convenience. When masking a small number of tokens, this method does not cause significant issues because the sum of attention weights is close to 1. However, when masking a large number of tokens, the sum of attention weights is not preserved, leading to model impairment and outputting [UNK] (Unknown) tokens. Consequently, the model cannot respond, resulting in an F1 Score of 0.
>
> Since our intention was to prevent the model from seeing specific tokens, not to impair the model, we re-ran the experiment using the former method, masking before the softmax. The former method can preserve the sum of attention weights, thereby preventing model impairment. We have replaced the results in Fig 4(a) with the new results and described the method in Appendix C.3. With this change, the model maintains about 50% F1 Score.

---

> > ### Author Response · Authors · 2024-11-23
> > **Response to Reviewer P74f (2/3)**
> >
> > ## C. Interpretation of the Results in Figure 4(a) (revised as Fig 3(b))
> >
> > Based on our understanding, the reviewer's concern is about why the performance difference between the visual sink token pruning and the random token pruning strategies persists when $\tau < 10$. We believe that the concern is due to the contradictory results (i.e., F1 Score of 0) and the reviewer's intuition is indeed correct. As we have replaced the experiment as stated in B, we provide a more detailed explanation for the revised experimental results below.
> >
> > As shown in Fig 3(a), when $\tau$ decreases to 15, the standard visual tokens start to be pruned. However, most standard visual tokens are still not pruned and visual sink tokens still dominate among the pruned tokens. Therefore, the performance is maintained compared to the random visual token pruning. The difference in performance becomes smaller as $\tau$ decreases, as more standard visual tokens are pruned. We have clarified the explanations of experimental results.
> >
> > [1] Cao, Shuyang, and Lu Wang. "Attention Head Masking for Inference Time Content Selection in Abstractive Summarization." Proceedings of the 2021 Conference of the North American Chapter of the Association for Computational Linguistics: Human Language Technologies. 2021.
> >
> > [2] Geva, Mor, et al. "Dissecting Recall of Factual Associations in Auto-Regressive Language Models." Proceedings of the 2023 Conference on Empirical Methods in Natural Language Processing. 2023.
> >
> > [3] Neo, Clement, et al. "Towards Interpreting Visual Information Processing in Vision-Language Models." 2024.
> >
> > [4] Wang, Lean, et al. "Label Words are Anchors: An Information Flow Perspective for Understanding In-Context Learning." Proceedings of the 2023 Conference on Empirical Methods in Natural Language Processing. 2023.
> >
> > [5] Mohebbi, Hosein, et al. "Quantifying Context Mixing in Transformers." Proceedings of the 17th Conference of the European Chapter of the Association for Computational Linguistics. 2023.
> >
> > [6] Jin, Zhuoran, et al. "Cutting Off the Head Ends the Conflict: A Mechanism for Interpreting and Mitigating Knowledge Conflicts in Language Models." Findings of the Association for Computational Linguistics. 2024.
> >
> > # W2 / Q3. Ambiguity in Scope of Token Selection for Attention Redistribution Strategy
> >
> > > W2. The scope of the attention redistribution strategy remains ambiguous. Specifically, which tokens are targeted by this strategy? In practice, attention redistribution could apply to both input instruction tokens and output tokens, but the implementation should align with the analytical findings. If the strategy is applied only to specific tokens, it is crucial to specify how these tokens are selected; if applied to all output tokens, the analysis should demonstrate stable consistency in the attention assigned to sink tokens. However, this consistency is not currently evident in the analysis.
> >
> > > Q3. Regarding the selection of image-centric attention heads, is the VAR strategy applied to all text tokens, or only to specific text tokens, or only to output tokens? If the experiment aligns with the analysis that VAR strategy only applied to specific object token, like "clock" in "Is there a clock in this image?", how are these particular tokens identified? If applied to all text tokens, is the visual token sink phenomenon consistent across all text tokens, which seems unlikely?
> >
> > We apologize for the lack of clarity in the token selection for the attention redistribution strategy. The target tokens of the VAR strategy are **all text tokens**, including both input instruction tokens and output tokens. We have added Fig 13, which visualizes the visual attention maps between all text tokens and visual tokens, to demonstrate that the visual attention sink is consistent across all text tokens. Therefore, VAR does not have the post-processing step to identify specific tokens. As the text tokens related to the visual information have more image-centric heads (Appendix A.3), the VAR strategy is automatically applied more effectively to these tokens.
> >
> > Meanwhile, in most of the figures, we only show the text-to-image attention map for specific object tokens to understand the concept more intuitively. However, we acknowledge that this may cause confusion about the scope of VAR. We have added clarifications in the caption of Fig 1 and the main text in Sec 5 to emphasize that the VAR strategy is applied to all text tokens.

---

> > > ### Author Response · Authors · 2024-11-23
> > > **Response to Reviewer P74f (3/3)**
> > >
> > > # Q4. Inference Process of Attention Redistribution
> > >
> > > > Q4. During inference, is the attention redistribution performed layer by layer? Specifically, does the forward pass first obtain original results, identify image-centric attention heads, then update attention weights, and proceed with attention weight redistribution in the next layer?
> > >
> > > Yes, the attention redistribution is performed layer by layer. A few more steps are added to the calculation of attention weights for each layer to identify image-centric attention heads and redistribute attention weights.

---

> > > ### Comment · Reviewer_P74f · 2024-11-27
> > > **Response to Authors (1/2)**
> > >
> > > I believe the author has successfully addressed Q1 and Q4; however, the critical issue of Q2 remains unresolved, and there is still some confusion regarding Q3.
> > >
> > > ## **Q2. Detailed Explanation of Figure 3 (Previously Figure 4)**
> > >
> > > The author’s explanation of the initial experimental results is quite surprising. I had speculated whether the author might have adjusted the attention scores *after* the softmax operation. If this were the case, **it would essentially disrupt the input distribution of the model entirely**, particularly given the high attention scores of certain tokens. It is difficult to believe such adjustments would have no effect on the results.
> > >
> > > It’s worth mentioning that I am not familiar with the methods described in [1][2], but I really familiar with [3]. In that study, the authors modified the attention weights *before* the softmax operation rather than afterward.
> > >
> > > Additionally, regardless of whether we refer to the updated or original version of the paper, the results in Figure 3 (previously Figure 4) remain puzzling. In Figure 3(a), we clearly observe **many tokens with $\tau > 40$**, while the range $10 < \tau < 20$ contains relatively **few tokens**. If, as the author claims, the performance drop in Figure 3(b) occurs because normal visual tokens are pruned when $\tau < 15$, leading to significant degradation, then pruning random visual tokens when $\tau > 40$ should presumably **result in even more normal visual tokens being pruned**. However, at this stage, the performance degradation is **smaller** than when **pruning visual sink tokens** in the range $10 < \tau < 20$. How is this possible?
> > >
> > > To be frank, most other aspects of this work appear quite solid. However, the results presented in Figure 3 significantly undermine my confidence in the experimental findings. Unfortunately, even with the updated results, my concerns regarding this issue remain unresolved.
> > >
> > > Given the extension of the discussion period, I will temporarily maintain my current rating while awaiting the author’s further response to Q2. I have noticed that the other reviewers have no objections to the results in Figure 3, and I welcome any discussions on this point with the other reviewers. If I have indeed misunderstood these results, I sincerely apologize in advance.
> > >
> > > [1] Mohebbi, Hosein, et al. "Quantifying Context Mixing in Transformers." Proceedings of the 17th Conference of the European Chapter of the Association for Computational Linguistics. 2023.
> > > [2] Jin, Zhuoran, et al. "Cutting Off the Head Ends the Conflict: A Mechanism for Interpreting and Mitigating Knowledge Conflicts in Language Models." Findings of the Association for Computational Linguistics. 2024.
> > > [3] Wang, Lean, et al. "Label Words are Anchors: An Information Flow Perspective for Understanding In-Context Learning." Proceedings of the 2023 Conference on Empirical Methods in Natural Language Processing. 2023.

---

> ### Comment · Reviewer_P74f · 2024-11-27
> **Response to Authors (2/2)**
>
> ## **Q3. Stability Characteristics of Visual Sink Tokens**
>
> I sincerely appreciate the author’s clarification regarding the scope of attention redistribution and the results presented in Figure 13, which demonstrate that sink tokens are consistently assigned higher attention scores. This has resolved most of my confusion. However, I still find it somewhat surprising if this method is applied to all text tokens.
>
> In the proposed method, the visual sink tokens at each layer are dynamic. However, for each output metric, the corresponding sink tokens remain constant. I would like to confirm one last time the conclusion the author is attempting to draw: *“For a single input sample, the sink tokens corresponding to all output tokens remain constant, while they are dynamic across layers.”* Is this correct?

---

> ### Author Response · Authors · 2024-11-30
> **Further Response to Reviewer P74f (1/3)**
>
> We sincerely appreciate the reviewer's insightful and intriguing questions. We welcome the opportunity to address these concerns and engage in further discussions about our research. Below, we provide detailed responses to each question. For clarity, we address Q3 first, followed by Q2.
>
> # Response to Q3. Stability Characteristics of Visual Sink Tokens
>
> > In the proposed method, the visual sink tokens at each layer are dynamic. However, for each output metric, the corresponding sink tokens remain constant. I would like to confirm one last time the conclusion the author is attempting to draw: “*For a single input sample, the sink tokens corresponding to all output tokens remain constant, while they are dynamic across layers.*” Is this correct?
>
> Yes, the reviewer's interpretation is indeed accurate. Sink tokens are dynamic across layers but remain constant within a single layer. We would like to clarify one additional point: as shown in Figure 8 and discussed in Appendix A.2, sink tokens emerge in the early layers and typically persist until the final layer. Therefore, the term "dynamic" does not imply frequent changes from layer to layer; rather, sink tokens rarely change once they emerge in the early layers. We hope this clarification resolves the confusion.

---

> > ### Author Response · Authors · 2024-11-30
> > **Further Response to Reviewer P74f (2/3)**
> >
> > # Response to Q2. Detailed Explanation of Figure 3
> >
> > > The author’s explanation of the initial experimental results is quite surprising. I had speculated whether the author might have adjusted the attention scores *after* the softmax operation. If this were the case, **it would essentially disrupt the input distribution of the model entirely**, particularly given the high attention scores of certain tokens. It is difficult to believe such adjustments would have no effect on the results.
> >
> > Based on our understanding, the reviewer is concerned about how the model's performance remains unaffected despite the attention weights of the visual sink token being set to zero *after* the softmax operation. If our understanding is correct, we recognize that the reviewer might think this way, and we acknowledge that it could seem somewhat unintuitive. There are two reasons why the model's performance remains stable despite the zeroed attention weights.
> >
> > ### 1) Sink tokens have high attention weights, but small vector norms
> >
> > Previous studies found that sink tokens have high attention weights but small vector norms [1, 2]. The multi-head attention (MHA) mechanism in transformers can be expressed as follows:
> >
> > $$
> > \text{MHA}^{\ell, h} (\boldsymbol{x}^{\ell-1} _ i) = \sum _ {j \leq i} \alpha^{\ell, h} _ {i, j} \boldsymbol{x} _ j^{\ell-1} \boldsymbol{W} _ {OV} ^ {\ell, h}.
> > $$
> >
> > Although the attention weights of the sink tokens (i.e., $\alpha ^ {\ell, h} _ {i, j} ~ (i \in \mathcal{I} _ {\textsf{txt}}, j \in \check{\mathcal{I}} ^ \ell _ \textsf{vis})$) are high, the vector norms of the sink tokens (i.e., $\Vert \boldsymbol{x} _ j ^ {\ell-1} \boldsymbol{W} _ {OV} ^ {\ell, h} \Vert$) are small. We compare the final contributions of the visual sink tokens $\Vert \alpha ^ {\ell, h} _ {i, j} \boldsymbol{x} _ j ^ {\ell-1} \boldsymbol{W} _ {OV} ^ {\ell, h} \Vert$ ($j \in \check{\mathcal{I}} ^ \ell _ \textsf{vis}$) with those of the visual non-sink tokens ($j \in \mathcal{I} _ \textsf{vis} \setminus \check{\mathcal{I}}^\ell _ \textsf{vis}$) below.
> >
> > | Token Type | Visual Sink Tokens | Visual Non-Sink Tokens |
> > |:----------:|:------------------:|:----------------------:|
> > | Contribution $\Vert \alpha ^ {\ell, h} _ {i, j} \boldsymbol{x} _ j ^ {\ell-1} \boldsymbol{W} _ {OV} ^ {\ell, h} \Vert$ | $7.95 \times 10 ^ {-4}$ | $3.74 \times 10 ^ {-3}$ |
> >
> > The final contributions are calculated in 100 samples and averaged. As shown in the table, the final contributions of the visual sink tokens are smaller than those of the visual non-sink tokens. Therefore, even though the sink tokens have high attention weights, they have less influence compared to other visual tokens due to their small vector norms.
> >
> > ### 2) Transformer architecture is generally robust to ablations.
> >
> > Transformer architectures are known to be robust to various ablations, such as the removal of attention heads or layers [3, 4, 5]. The reasons for this robustness are not yet fully understood, but it is believed that transformers possess a highly redundant structure enabled by residual connections [4] and activate self-repair mechanisms when certain components are removed [6, 7]. Therefore, if visual sink tokens do not carry important information, the model can potentially compensate for the changes in hidden states caused by zeroing out the attention weights of the sink tokens.
> >
> > To summarize, the stability of the model's performance despite zeroing out the attention weights of the visual sink tokens can be attributed to the small vector norms of the sink tokens and the robustness of the transformer architecture.
> >
> > We have reuploaded the response to Q2 due to an implementation error in calculating the final contributions of visual sink and non-sink tokens. We apologize for the inconvenience.
> >
> > [1] Kobayashi, Goro, et al. "Attention is Not Only a Weight: Analyzing Transformers with Vector Norms." Proceedings of the 2020 Conference on Empirical Methods in Natural Language Processing. 2020.
> >
> > [2] Sun, Mingjie, et al. "Massive Activations in Large Language Models." Conference on Language Modeling (COLM), 2024.
> >
> > [3] Michel, Paul, Omer Levy, and Graham Neubig. "Are Sixteen Heads Really Better than One?" Advances in Neural Information Processing Systems. 2019.
> >
> > [4] He, Shwai, et al. "What Matters in Transformers? Not All Attention is Needed." 2024.
> >
> > [5] Lad, Vedang, Wes Gurnee, and Max Tegmark. "The Remarkable Robustness of LLMs: Stages of Inference?" 2024.
> >
> > [6] McGrath, Thomas, et al. "The Hydra Effect: Emergent Self-repair in Language Model Computations." 2023.
> >
> > [7] Rushing, Cody, and Neel Nanda. "Explorations of Self-Repair in Language Models." Proceedings of the 41st International Conference on Machine Learning. 2024.

---

> > > ### Comment · Reviewer_P74f · 2024-12-01
> > > **Further Response to Authors (1/3)**
> > >
> > > I believe the author has fully addressed Q3. However, the conclusions drawn from Q3 have further implications for the experimental setup in Q2, and the issues in Q2 still remain.
> > >
> > > ## **About setting attention weights to 0**
> > >
> > > I must admit that I am not very familiar with the characteristics of sink tokens, and I appreciate the author's response for providing me with substantial background knowledge on this topic. That said, I am still slightly puzzled. In [1], it was noted that setting the attention sink tokens proposed in the paper to zero after the softmax operation causes a significant drop in model performance. Similarly, in MLLMs, system tokens in deeper layers also exhibit characteristics of sink tokens, and pruning them likewise **leads to a dramatic performance decline.** My question is: do these two types of tokens align with the author’s definition of sink tokens? Do they exhibit the high activation values in specific dimensions as described? Furthermore, why does pruning them cause such a drastic drop in model performance?
> > >
> > > Additionally, regarding the operation of setting attention to zero in [2], I revisited the paper and found that the authors did not clarify whether the experiments in Section 2.2 involved setting the attention to zero before or after the softmax operation. Upon reviewing their open-source code, specifically the file `icl.analysis.shallow_layer.py`, it appears that they set the attention to zero **before** the softmax operation.
> > >
> > > Given the approaching deadline, **if the author has limited time, please feel free to temporarily disregard this question.** I believe resolving the other issue will help me adjust the rating to a positive score more quickly.
> > >
> > > [1] Xiao, Guangxuan, et al. "Efficient streaming language models with attention sinks." *arXiv preprint arXiv:2309.17453* (2023).
> > > [2] Wang, Lean, et al. "Label Words are Anchors: An Information Flow Perspective for Understanding In-Context Learning." Proceedings of the 2023 Conference on Empirical Methods in Natural Language Processing. 2023.

---

> ### Author Response · Authors · 2024-11-30
> **Further Response to Reviewer P74f (3/3)**
>
> > It’s worth mentioning that I am not familiar with the methods described in [1][2], but I really familiar with [3]. In that study, the authors modified the attention weights before the softmax operation rather than afterward.
>
> We appreciate the reviewer for pointing it out. To the best of our knowledge, in [3], the authors set the attention weights to zero *after* the softmax operation in the analysis section (Section 2.2) and propose a attention re-weighting strategy by modifying the attention weights *before* the softmax operation in the application section (Section 3.1). If we have misunderstood the paper, we sincerely apologize for the confusion.
>
> [1] Mohebbi, Hosein, et al. "Quantifying Context Mixing in Transformers." Proceedings of the 17th Conference of the European Chapter of the Association for Computational Linguistics. 2023.
>
> [2] Jin, Zhuoran, et al. "Cutting Off the Head Ends the Conflict: A Mechanism for Interpreting and Mitigating Knowledge Conflicts in Language Models." Findings of the Association for Computational Linguistics. 2024.
>
> [3] Wang, Lean, et al. "Label Words are Anchors: An Information Flow Perspective for Understanding In-Context Learning." Proceedings of the 2023 Conference on Empirical Methods in Natural Language Processing. 2023.
>
> > Additionally, regardless of whether we refer to the updated or original version of the paper, the results in Figure 3 (previously Figure 4) remain puzzling. In Figure 3(a), we clearly observe many tokens with $\tau > 40$, while the range $10 < \tau < 20$ contains relatively few tokens. If, as the author claims, the performance drop in Figure 3(b) occurs because normal visual tokens are pruned when $\tau < 15$, leading to significant degradation, then pruning random visual tokens when $\tau > 40$ should presumably result in even more normal visual tokens being pruned. However, at this stage, the performance degradation is smaller than when pruning visual sink tokens in the range $10 < \tau < 20$. How is this possible?
>
> We are truly grateful for this insightful question, and we appreciate the opportunity to address it. The reason for the puzzling results in Figure 3 is that "random visual tokens" experiment is more robust to the token masking process than the "visual sink tokens" experiment.
>
> First, we would like to provide a detailed description of the experimental design to explain it. For each layer $\ell$, we select the visual tokens that satisfy the condition $\phi (\boldsymbol{x} ^{\ell-1} _ j) \geq \tau$ and mask them (depicted as "visual sink tokens" in Figure 3(b)). Similarly, we randomly select the same number of visual tokens and mask them (depicted as "random visual tokens" in Figure 3(b)). In the "visual sink tokens" experiment, although we select visual tokens for each layer, the selected tokens are almost consistent across layers because the hidden states do not change significantly from layer to layer (due to the residual connections). In contrast, in the "random visual tokens" experiment, the selected tokens differ for each layer because the selection is random. If the masked token contains important information, the model has almost no chance of acquiring that information in the "visual sink tokens" experiment. However, in the "random visual tokens" experiment, the model can recover the information from other layers.
>
> This difference in the robustness of the two experiments causes the "random visual tokens" experiment to maintain higher performance than the "visual sink tokens" experiment, even when a larger number of normal visual tokens are masked. Additionally, since random visual tokens also include visual sink tokens, there is a possibility of selecting visual sink tokens in the "random visual tokens" experiment. Therefore, directly comparing the two graphs at different $\tau$ values may be challenging. Instead, we would like to suggest considering the "random visual tokens" experiment as a reference for the "visual sink tokens" experiment at the same $\tau$ value.
>
> We sincerely hope that our responses have helped to clarify any remaining uncertainties. If we have misinterpreted any aspect of the reviewer's question, please notify us, and we will quickly resolve it. Lastly, we would like to express our gratitude to the reviewer for their valuable feedback and time spent reviewing our work.

---

> > ### Comment · Reviewer_P74f · 2024-12-01
> > **Further Response to Authors (2/3)**
> >
> > ## **About proportion of performance degradation**
> >
> > OK, I believe this is the most critical part. If I have correctly understood the author’s explanation, the claim is that during the pruning of random visual tokens, some visual sink tokens are also included. As a result, when $\phi > 40$, the number of pruned normal visual tokens is not significantly higher than the number pruned in the visual sink tokens experiment when $10 < \phi < 20$.
> >
> > However, I had already considered the efficiency of pruning when raising this question. Since I lack access to specific data, I could only infer from Figure 3(a) that the ratio of normal visual tokens to visual sink tokens is approximately 4:1, while the ratio of tokens with $10 < \phi < 20$ to those with $\phi > 40$ is at least 5:1. Therefore, in the random visual tokens experiment, when $\phi > 40$, the number of pruned normal visual tokens should be at least **four times** the number pruned in the visual sink tokens experiment when $10 < \phi < 20$.
> >
> > Yet, in the experiment, both scenarios show the **same proportion of performance degradation**, particularly when $\phi > 45$, where the degradation becomes inexplicably low. This inconsistency is quite puzzling to me.

---

> > ### Comment · Reviewer_P74f · 2024-12-01
> > **Further Response to Authors (3/3)**
> >
> > ## **About fixed position of visual tokens pruning**
> >
> > Additionally, the author’s response raises a new question. The author mentioned that **in deeper layers, visual sink tokens are nearly fixed**, which means pruning visual tokens in these layers is also fixed. However, in the random visual tokens experiment, the pruned visual tokens vary dynamically across layers. Does this discrepancy make the comparison unfair, potentially leading to misleading results? **Pruning tokens from fixed positions in each layer versus dynamically pruning tokens from different positions inherently causes different levels of disruption to the information flow.**
> >
> > More specifically, if visual tokens are pruned from fixed positions in each layer, wouldn’t the impact on model performance be more limited? I conducted a simple experiment myself: **I randomly selected 25% of the visual tokens and pruned these same fixed positions across all layers.** The resulting POPE test score was **82.4**, which seems comparable to the results shown in the visual sink tokens experiment in the figure-3(b).
> >
> > Does this suggest that the observed performance degradation might not stem from differences between sink tokens and normal tokens, **but rather from the distinction between pruning fixed versus dynamic positions across layers**?

---

> > > ### Author Response · Authors · 2024-12-02
> > > **Further response to reviewer p74f Ⅱ (1/3)**
> > >
> > > We thank you for taking the time to provide us with additional questions. We have addressed them below.
> > >
> > > # Response to #1: About setting attention weights to 0
> > >
> > > > In [1], it was noted that setting the attention sink tokens proposed in the paper to zero *after* the softmax operation causes a significant drop in model performance. Similarly, in MLLMs, system tokens in deeper layers also exhibit characteristics of sink tokens, and pruning them likewise **leads to a dramatic performance decline**. My question is: do these two types of tokens align with the author’s definition of sink tokens? Do they exhibit the high activation values in specific dimensions as described? Furthermore, why does pruning them cause such a drastic drop in model performance?
> > >
> > > We appreciate the insightful question about attention sinks. The initial token of LLMs (the `<bos>` token) and certain system tokens in MLLMs (including the `<bos>` token) align with our definition of sink tokens, exhibiting high activation values in sink dimensions. A significant performance drop, as reported in [1], is observed when the initial sink tokens (particularly the `<bos>` token) are excluded from the attention window. To clarify, this scenario is technically equivalent to masking the attention weights of the sink tokens before the softmax operation, as the sink tokens are not factored into the softmax calculation.
> > >
> > > Notably, this performance decline is a distinct property of the `<bos>` token. As shown in Figure 12 of [1], the `<bos>` token has exceptionally high attention weights (ranging from 0.4 to 1.0). Consequently, masking the `<bos>` token significantly alters the attention distribution, leading to a substantial performance drop. In contrast, other sink tokens, including visual sink tokens in MLLMs, do not exhibit such high attention weights like <bos> token (though they still have higher attention weights compared to other tokens). As a result, pruning these tokens causes minimal changes to the attention distribution, and the associated performance drop is negligible compared to that of the `<bos>` token.
> > >
> > > > Additionally, regarding the operation of setting attention to zero in [2], I revisited the paper and found that the authors did not clarify whether the experiments in Section 2.2 involved setting the attention to zero before or after the softmax operation. Upon reviewing their open-source code, specifically the file `icl.analysis.shallow_layer.py`, it appears that they set the attention to zero **before** the softmax operation.
> > >
> > > We apologize for the misunderstanding regarding the attention zeroing operation in [2]. Thank you for pointing out and clarifying the implementation details.
> > >
> > >
> > > [1] Xiao, Guangxuan, et al. "Efficient streaming language models with attention sinks." arXiv preprint arXiv:2309.17453 (2023).
> > >
> > > [2] Wang, Lean, et al. "Label Words are Anchors: An Information Flow Perspective for Understanding In-Context Learning." Proceedings of the 2023 Conference on Empirical Methods in Natural Language Processing. 2023.

---

> ### Author Response · Authors · 2024-12-02
> **Further response to reviewer p74f Ⅱ (2/3)**
>
> # Response to #2: About proportion of performance degradation
>
> > If I have correctly understood the author’s explanation, the claim is that during the pruning of random visual tokens, some visual sink tokens are also included. As a result, when
> $\phi > 40$, the number of pruned normal visual tokens is not significantly higher than the number pruned in the visual sink tokens experiment when $10 < \phi < 20$.
>
> > However, I had already considered the efficiency of pruning when raising this question. Since I lack access to specific data, I could only infer from Figure 3(a) that the ratio of normal visual tokens to visual sink tokens is approximately 4:1, while the ratio of tokens with $10 < \phi  < 20$ to those with $\phi > 40$ is at least 5:1. Therefore, in the random visual tokens experiment, when $\phi  > 40$, the number of pruned normal visual tokens should be at least **four times** the number pruned in the visual sink tokens experiment when $10 < \phi < 20$.
>
> > Yet, in the experiment, both scenarios show the **same proportion of performance degradation**, particularly when $\phi > 45$, where the degradation becomes inexplicably low. This inconsistency is quite puzzling to me.
>
> We are sorry for any confusion caused by our explanation. Our primary claim in "Further Response to Reviewer P74f (3/3)" was that masking the same fixed tokens across layers leads to greater performance degradation compared to masking random tokens per layer, even when a larger proportion of normal visual tokens is pruned in the latter case. (We will discuss this in more detail in the next section.) We acknowledge that our additional claim regarding the potential selection of visual sink tokens in the "random visual tokens" experiment is not the primary reason why the performance degradation does not become significantly higher when $\phi (x) > 40$. Indeed, when $\phi (x) > 40$, the number of masked normal visual tokens is approximately **three times** the number of masked visual sink tokens for $10 < \phi (x) < 20$, which aligns closely with the ratio inferred by the reviewer. We would like to explain why the performance degradation does not increase significantly when $\phi (x) > 40$ in the next section.

---

> ### Author Response · Authors · 2024-12-02
> **Further response to reviewer p74f Ⅱ (3/3)**
>
> # Response to #3: About fixed position of visual tokens pruning
>
> > Additionally, the author’s response raises a new question. The author mentioned that **in deeper layers, visual sink tokens are nearly fixed**, which means pruning visual tokens in these layers is also fixed. However, in the random visual tokens experiment, the pruned visual tokens vary dynamically across layers. Does this discrepancy make the comparison unfair, potentially leading to misleading results? **Pruning tokens from fixed positions in each layer versus dynamically pruning tokens from different positions inherently causes different levels of disruption to the information flow.**
>
> > More specifically, if visual tokens are pruned from fixed positions in each layer, wouldn’t the impact on model performance be more limited?
>
> > Does this suggest that the observed performance degradation might not stem from differences between sink tokens and normal tokens, **but rather from the distinction between pruning fixed versus dynamic positions across layers**?
>
> We appreciate the insightful question about the impact of fixed versus dynamic masking on model performance. We conducted additional experiments to investigate the impact of fixed versus dynamic masking on model performance. Specifically, we compared the performance degradation of fixed masking and dynamic masking. The number of tokens masked in these experiments is the same as the number of tokens masked in the “random visual tokens” experiment ($\phi (x) > 40$ and $\phi (x) > 45$). Thus, the dynamic masking experiment is expected to show performance degradation similar to the our “random visual tokens” experiment in Figure 3(b). Since the discussion period is limited, we evaluated the F1 score in about 10% of the POPE benchmark. The results are shown below:
>
> | $\phi (x)$ | Fixed Masking | Dynamic Masking |
> |--------|---------------|-----------------|
> | $ >40 $ 	| 52.0      	| 61.3        	|
> | $ >45 $	| 61.1      	| 77.6        	|
>
> The results of the dynamic masking experiment are consistent with the “random visual tokens” experiment in Figure 3(b). These results show that fixed masking has a greater impact on performance degradation than dynamic masking. Therefore, the comparison between the "visual sink tokens" experiment (similar to Fixed Masking) and the "random visual tokens" experiment (similar to Dynamic Masking) is biased in a way that disadvantages the "visual sink tokens" experiment. Even so, the sink token experiment consistently maintains performance compared to the “random visual tokens” experiment for $\tau > 20$. This result supports that the visual sink tokens are meaningless.
>
> These results also align with the discussion in #2. Since the number of masked normal visual tokens for $\phi (x) > 40$ is approximately three times the number of masked normal visual tokens for $10 < \phi (x) < 20$, the performance degradation in fixed masking at $\tau = 40$ is higher than that of $\tau = 10$ in the "visual sink tokens" experiment. This holds true for fixed masking at $\tau = 45$ and $\tau = 15$ in the "visual sink tokens" experiment as well.
>
> > I conducted a simple experiment myself: **I randomly selected 25% of the visual tokens and pruned these same fixed positions across all layers**. The resulting POPE test score was **82.4**, which seems comparable to the results shown in the visual sink tokens experiment in the figure-3(b).
>
> We appreciate the reviewer's effort in conducting the experiment. After performing our own "Fixed Masking" experiments, we are questioning the discrepancy between the reviewer's results and our own. Since we do not have complete details about the reviewer's experiment, we cannot provide a definitive explanation. However, we believe that the discrepancy may stem from the difference between pruning and masking. Pruning involves excluding visual tokens before they enter the layer, while masking is a post-processing step that eliminates their impact after the attention calculation has been completed within the layer. While pruning maintains performance even when a large number of tokens are removed [1], masking leads to performance degradation even when only a small number of tokens are excluded [2, 3]. If we have misunderstood the reviewer's experiment, we sincerely apologize and would appreciate it if the reviewer could provide more details about the experiment.
>
> [1] Chen, Liang, et al. "An Image is Worth 1/2 Tokens After Layer 2: Plug-and-Play Inference Acceleration for Large Vision-Language Models." European Conference on Computer Vision. 2024.
>
> [2] Geva, Mor, et al. "Dissecting Recall of Factual Associations in Auto-Regressive Language Models." Proceedings of the 2023 Conference on Empirical Methods in Natural Language Processing. 2023.
>
> [3] Neo, Clement, et al. "Towards Interpreting Visual Information Processing in Vision-Language Models." 2024.

---

> ### Comment · Reviewer_P74f · 2024-12-03
> **Final Response to Authors (1/2)**
>
> ## **1. About the Fixed Position of Visual Tokens Pruning**
>
> In my expectation, **fixed masking** should have a smaller impact on the model's performance compared to **dynamic masking**. As a result, I hypothesized that the experimental results for the *visual sink tokens* scenario would show a relatively smaller performance degradation, meaning that the blue line in Figure 3(b) should shift upward. However, the author's experiments suggest the opposite: fixed masking has a more significant negative impact on the model's performance than dynamic masking, which implies that the blue line in Figure 3(b) would shift further downward.
>
> Unfortunately, my own experimental results do not support this claim. My experiments indicate that pruning 25% of visual tokens at fixed positions does not lead to significant performance degradation. In my implementation, I modified the `attention_mask` parameter in the `llama_model` module to mask out a portion of the visual tokens. Furthermore, in the author's experimental results, masking only approximately 15% of tokens resulted in a performance drop close to 50%, which is nearly equivalent to random guessing. Could the authors provide more details about their experimental setup?

---

> > ### Comment · Reviewer_P74f · 2024-12-03
> > **Final Response to Authors (2/2)**
> >
> > ## **2. About the Proportion of Performance Degradation**
> >
> > Regrettably, the author's response did not resolve my confusion on this issue. If my understanding is correct, the authors aim to demonstrate that in the *random visual tokens* experiment, when $\phi(x) > 40$, pruning visual tokens at fixed positions results in a more significant performance degradation than in the *visual sink tokens* experiment, where $10 < \phi(x) < 20$. However, in the *visual sink tokens* experiment, when $10 < \phi(x) < 20$, the pruned tokens should already correspond to **normal visual tokens**, which should effectively equate to dynamic pruning. Therefore, the authors need to provide an explanation for the inconsistency in the degradation rates presented in Figure 3 (b). This discrepancy cannot be sufficiently accounted for by the results of pruning at fixed positions, especially considering that there appears to be a significant difference between our experimental results for fixed-position pruning.

---

> > > ### Comment · Reviewer_P74f · 2024-12-03
> > > **Final Evaluation Decision**
> > >
> > > I kindly request the AC to carefully review both my latest response and the author's final reply, as significant disagreements appear to persist until the very end. Specifically, I find the current explanation regarding Figure 3(b) and the results of fixed position experiments in *Further response to reviewer p74f Ⅱ (3/3)* difficult to accept. Consequently, I must maintain my current rating. If there has indeed been a misunderstanding, I sincerely apologize in advance.

---

> > > > ### Author Response · Authors · 2024-12-04
> > > > **Final Response to Reviewer P74f**
> > > >
> > > > First of all, we would like to express our sincere gratitude to the reviewer for investing time and effort in providing detailed feedback on our work. As this is our final response, we have also included a general response, particularly addressing the discussions between the reviewer and us. We kindly ask the reviewer to refer to the general response, as it provides additional context for the discussion. Below, we address the reviewer’s specific comments.
> > > >
> > > > > Unfortunately, my own experimental results do not support this claim. My experiments indicate that pruning 25% of visual tokens at fixed positions does not lead to significant performance degradation. In my implementation, I modified the `attention_mask` parameter in the `llama_model` module to mask out a portion of the visual tokens. Furthermore, in the author's experimental results, masking only approximately 15% of tokens resulted in a performance drop close to 50%, which is nearly equivalent to random guessing. Could the authors provide more details about their experimental setup?
> > > >
> > > > We appreciate the reviewer for sharing the details of their experiment. Despite our efforts, we were unable to replicate the reviewer's results because it was unclear which specific queries and keys were masked and at what positions. Appendix C.3 provides a detailed description of our experimental setup, specifically explaining how we masked the visual tokens (i.e., specific queries and keys). In the final version of the paper, we will further refine this description to offer a more intuitive understanding of the masking process.
> > > >
> > > > > Regrettably, the author's response did not resolve my confusion on this issue. If my understanding is correct, the authors aim to demonstrate that in the random visual tokens experiment, when $\phi(x) > 40$, pruning visual tokens at fixed positions results in a more significant performance degradation than in the visual sink tokens experiment, where $10 < \phi(x) < 20$. However, in the visual sink tokens experiment, when $10 < \phi(x) < 20$, the pruned tokens should already correspond to **normal visual tokens**, **which should effectively equate to dynamic pruning**. Therefore, the authors need to provide an explanation for the inconsistency in the degradation rates presented in Figure 3 (b). This discrepancy cannot be sufficiently accounted for by the results of pruning at fixed positions, especially considering that there appears to be a significant difference between our experimental results for fixed-position pruning.
> > > >
> > > > As mentioned in our previous response (Further Response to Reviewer P74f (3/3)), visual tokens—whether visual sink tokens or normal visual tokens—remain nearly consistent across layers because the hidden states do not change significantly from layer to layer (due to the redundancy from residual connections). Therefore, when $10 < \phi(x) < 20$ in the “visual sink token” experiment, the masked normal visual tokens are similar to the **fixed masking setting, not dynamic masking**. Given this, we believe that our explanation in "*Further Response to Reviewer P74f Ⅱ (3/3)*" sufficiently addresses the reviewer’s concern.
> > > >
> > > > It is unfortunate that we do not have more time for further discussion on this issue. However, we would like to emphasize that **the primary goal of the visual sink token experiment is to demonstrate that the visual sink token has a negligible impact on the model's performance**. This conclusion is supported by the results in Figure 3(b) and our discussion with the reviewers (further discussion can be found in the final general response). We hope that the reviewer finds our explanation satisfactory within the context of the entire paper.
> > > >
> > > > Finally, we sincerely thank the reviewer for their active participation in the discussion and for providing valuable insights throughout this process. It has been a meaningful and rewarding experience for us, and we truly appreciate it.

---

### Official Review · Reviewer_1YMG · 2024-11-05

**Soundness:** 3
**Presentation:** 2
**Contribution:** 3
**Rating:** 8
**Confidence:** 4

**Summary:**

This paper investigates "visual attention sinks" in LMMs. The authors pointed out that LMMs often allocate disproportionate attention to certain visual tokens, termed "visual sink tokens," regardless of their relevance to the corresponding text. This paper proposes Visual Attention Redistribution (VAR), a method that identifies and reallocates attention from these sink tokens to more pertinent visual information, enhancing model performance across various vision-language tasks. The authors demonstrate VAR's effectiveness without requiring additional model training.

**Strengths:**

1. The identification of visual attention sinks draws parallels with attention sinks in language models, providing a novel insight into the functioning of LMMs.
1. Overall well written and easy to follow. Fig 1 and Fig 2 provide a clear and strong motivation. Fig 3-5 also provide a step-by-step motivation and the design choices of the proposed method.
1. The method improves performance across multiple benchmarks, including general vision-language tasks, visual hallucination tasks, and vision-centric tasks.
1. The paper evaluates VAR on a wide range of benchmarks and conducts a series of quantitative and qualitative analyses, validating the effectiveness of the proposed VAR method.

**Weaknesses:**

1. The reviewer's biggest concern is about the determination of hyperparameter $\rho$. According to the paper and Fig 7(b), it is benchmark-dependent. Furthermore, a poorly chosen $\rho$ can lead to performance worse than the baseline performance. This parameter greatly limited the applicability and generalizability of the proposed method.

1. The tuning of $\rho$ on the benchmarks to find the best value is problematic and is against the principle of machine learning.

**Questions:**

1. Do newer LMMs with anyres and with visual encoder tuned jointly also have the characteristic of visual attention sinks? Do the observation and analyses generalize across LMMs with different language models and sizes?

1. In Fig 8, it looks like LLaVA + VAR has more visual sink tokens qualitatively.

1. Please include the reference of LLaVA-1.5-HD-13B.

---

> ### Author Response · Authors · 2024-11-23
> **Response to Reviewer 1YMG (1/2)**
>
> We thank the reviewer for the constructive review and insightful comments. We provide a detailed response to the reviewer's comments below. If there are any remaining concerns, please inform us, and we will address them promptly.
>
> # W1. Determination of Hyperparameter $\rho$
>
> > W1. The reviewer's biggest concern is about the determination of hyperparameter $\rho$. According to the paper and Fig 7(b), it is benchmark-dependent. Furthermore, a poorly chosen $\rho$ can lead to performance worse than the baseline performance. This parameter greatly limited the applicability and generalizability of the proposed method.
>
> (**Benchmark-dependent $\rho$**) We appreciate the reviewer for raising this question. To clarify, we use the same $\rho$ value for all the benchmarks in the same task type (0.8 for general vision-language task, 0.5 for visual hallucination task, and 0.9 for vision-centric task). In the paper, we use the terms 'task' and 'benchmark' differently. A 'task' refers to the type of problem (e.g., general vision-language task, visual hallucination task, and vision-centric task) and a 'benchmark' refers to a specific dataset (e.g., VQAv2, GQA, LLaVA-W, MM-Vet). A single $\rho$ value can improve the performance of various LMMs on all benchmarks for the same task type. The purpose of presenting Fig 7(b) (revised as 6(b)) is to obtain the plausible $\rho$ value for all the benchmarks in the same task type using only a single benchmark per task. We will further explain the rationale and justification for the choices of $\rho$ in W2.
>
> (**Worse performance with poorly chosen $\rho$**) We acknowledge that a poorly chosen $\rho$ can lead to performance worse than the baseline performance. However, in the reasonable range of $\rho$ values, the proposed method robustly improves the performance of various LMMs on various benchmarks. For example, if $\rho \geq 0.5$, VAR can consistently improve the performance on MME (general vision-language task), POPE (visual hallucination task), and CV-Bench (vision-centric task). Therefore, we argue that, to some extent, the proposed method remains applicable across various benchmarks within plausible ranges of $\rho$ values. We have discussed the hyperparameter $\rho$ more thoroughly in Sec 6.3. and Appendix B.2.
>
> # W2. Tuning of Hyperparameter $\rho$
>
> > W2. The tuning of $\rho$ on the benchmarks to find the best value is problematic and is against the principle of machine learning.
>
> Since there is no train/validation set for most of the benchmarks, we alternatively determine $\rho$ based on the single benchmark per task and use the determined $\rho$ for all the benchmarks in the same task type. To validate that the tuning process of $\rho$ is not benchmark-sensitive, we conducted two additional experiments. First, we used a “pseudo-validation set” by randomly sampling 10% of the samples from the benchmark and determined $\rho$ using the partial samples. Second, we applied another benchmark in the same task type to determine $\rho$. The results indicate that the same $\rho$ value can be obtained from partial samples of the benchmark or another benchmark in the same task type. The results indicate that we can find the applicable $\rho$ value with minimal tuning. We added the clarification about the tuning of $\rho$ in the manuscript (Sec 6.3. & Appendix B.2.) and include the results of the experiments with 10% of the samples in the benchmark (Appendix B.2.). The results are also provided here for ease of reference.
>
> | $\rho$ | 0 | 0.1 | 0.2 | 0.3 | 0.4 | 0.5 | 0.6 | 0.7 | 0.8 | 0.9 | 1 (baseline) |
> |---|---|---|---|---|---|---|---|---|---|---|---|
> | **General Vision-Language Task** | | | | | | | | | | | |
> | MME (10%) | 144.00 | 144.27 | 145.82 | 147.28 | 149.33 | 151.33 | 151.58 | 151.85 | 152.59 | 152.20 | 148.50 |
> | TextVQA (10%) | 57.9987 | 58.1770 | 58.2403 | 58.2501 | 58.2530 | 58.3042 | 58.3898 | 58.5275 | 58.6328 | 58.5320 | 58.1200 |
> | **Visual Hallucination Task** | | | | | | | | | | | |
> | POPE (10%) | 85.9012 | 85.9012 | 86.0124 | 86.1002 | 86.4020 | 86.5283 | 86.3606 | 86.3247 | 86.2029 | 85.9930 | 85.8700 |
> | 1-CHAIRs (10%) | 54.9705 | 55.0330 | 55.3408 | 56.1380 | 56.3189 | 56.5543 | 56.2280 | 55.5402 | 55.0523 | 54.7548 | 54.7021 |
> | **Vision-centric Task** | | | | | | | | | | | |
> | CV-Bench-2D (10%) | 56.0034 | 56.0146 | 56.8823 | 56.8842 | 56.9140 | 56.9430 | 57.0327 | 57.1328 | 57.5475 | 57.6000 | 56.1300 |
> | CV-Bench-3D (10%) | 58.2923 | 58.5538 | 58.5703 | 58.6111 | 58.6550 | 58.7530 | 58.7962 | 58.9041 | 58.9542 | 59.0000 | 58.2800 |

---

> ### Author Response · Authors · 2024-11-23
> **Response to Reviewer 1YMG (2/2)**
>
> # Q1. More Experiments with Various LMMs
>
> > Q1. Do newer LMMs with anyres and with visual encoder tuned jointly also have the characteristic of visual attention sinks? Do the observation and analyses generalize across LMMs with different language models and sizes?
>
> We have added two more LMMs, Qwen2-VL [1] and InternVL2 [2], which are more recent. Qwen2-VL is a anyres model with another language model (Qwen), and InternVL2 is a anyres model with a visual encoder tuned jointly. We observed that the visual attention sinks are also observed in these models. Vision encoder tuning may not affect the visual attention sink phenomenon, as the emergence of visual attention sinks is an intrinsic property of LLMs. We have added the investigation results about the hidden states of these models in the Appendix A.1. Also, we have conducted the same experiments with the proposed method on these models and it consistently improves the performance on these models. The results indicate that visual attention sink works well on various LMMs. We have added the results of the experiments on these models in the main paper (Table 1, 2, 3).
>
> [1] Wang, Peng, et al. "Qwen2-VL: Enhancing Vision-Language Model's Perception of the World at Any Resolution." 2024.
>
> [2] OpenGVLab Team. "InternVL2: Better than the Best—Expanding Performance Boundaries of Open-Source Multimodal Models with the Progressive Scaling Strategy". 2024.
>
> # Q2. Qualitative Results
>
> > Q2. In Fig 8, it looks like LLaVA + VAR has more visual sink tokens qualitatively.
>
> The background tokens in Fig 8 are not visual sink tokens, but noises. Although they have high attention weights, they do not have massive activation in specific dimensions and do not consistently appear in fixed locations. Each attention head does not attend perfectly to the relevant visual tokens, and some attention weights are allocated to the noise. Some may argue that our method potentially amplifies noise, which could adversely affect performance. However, the majority of the attention budget is allocated to important visual tokens, ensuring the effectiveness of our approach.
>
> # Q3. Citation
>
> > Q3. Please include the reference of LLaVA-1.5-HD-13B.
>
> We have added the reference of LLaVA-1.5-HD-13B in the manuscript. Thank you for pointing it out.

---

> > ### Comment · Reviewer_1YMG · 2024-11-25
> >
> > Thanks for the detailed analyses. The reviewer appreciates the efforts for the new experiments.
> >
> > The authors have addressed the reviewer's major concern about the hyperparameter $\rho$ by providing more experimental results. Specifically, they show that the value of $\rho$ is quite stable across benchmarks within each type of task. The tuning method of taking 10% of data as validation set also resolves the reviewer's concern of directly tuning on the benchmark.
> >
> > The authors also provide more experiments showing that the attention sink phenomenon can also be observed in newer LMMs with anyres as well as different families of LLMs.
> >
> > Overall, the authors have addressed the reviewer's major concerns. Based on the soundness and contribution of this paper, the reviewer recommend accepting this paper and has adjusted the score accordingly.

---

### Author Response · Authors · 2024-11-23
**General Response**

We appreciate all the reviewers for their constructive comments and insightful suggestions. Our research has been further refined and enriched thanks to their feedback. We have uploaded the revised manuscript, incorporating the reviewers' helpful comments. Below, we summarize the major changes made in the revised manuscript:

* More Recent LMMs for Evaluation: We have added two more LMMs, Qwen2-VL and InternVL2, which are more recent. The results show that VAR consistently improves the performance of these models, indicating that the proposed method is applicable to various LMMs. The results are presented in Table 1, 2, and 3. (Reviewer 1YMG, b7X3)
* More Discussions on Visual Attention Sink: We have discussed the characteristics of visual attention sinks, including token properties and emerging mechanisms, in more detail. We have also added the relationship between sink tokens in different models. Research on sink tokens in ViT [1] was incorporated into the related work (Section 2). The discussion is presented in Section 4.2 and Appendix A.2. (Reviewer KaLi)
* Reevaluating Token Masking Experiment in Section 4.2: Recognizing that the token masking experiment setup was misaligned with our intended objectives, we revised the setup and reevaluated the experiment. We also clarified the experimental settings and results. The updated results are presented in Figure 3(b), with further details available in Appendix C.3. (Reviewer P74f, b7X3)
* More Experiments and Discussions on Hyperparameters: We conducted additional experiments to validate the hyperparameter tuning process and demonstrate the robustness of VAR to hyperparameter choices. The results are summarized in Section 6.3 and detailed in Appendix B.2. (Reviewer 1YMG, KaLi)
* Validation of Visual Sink Tokens' Irrelevance to the Main Subject: We added quantitative validation of the trend showing that visual sink tokens are unrelated to the main subject of the image, using segmentation datasets, in Appendix A.2. (Reviewer b7X3)
* Clearer Explanations and Presentations:
  * Figure 3 and Figure 4 were integrated to provide a more coherent presentation of the experimental results.
  * The method for visualizing the attention maps in Figure 1 was clarified, and visual attention maps between all text tokens and visual tokens were included in Figure 13 to demonstrate the consistency of the visual attention sink across all text tokens. (Reviewer P74f)
  * The target tokens of VAR were explicitly detailed in Section 5. (Reviewer P74f)
  * Additional future research directions were included in Section 7. (Reviewer b7X3)

[1] Darcet, Timothée, et al. "Vision Transformers Need Registers." The Twelfth International Conference on Learning Representations. 2024.

---

### Author Response · Authors · 2024-12-04
**Final General Response (1/2)**

Dear AC and Reviewers,

We appreciate the time and effort the reviewers have dedicated to providing detailed feedback on our work. During the discussion period, we have received valuable suggestions, such as `(1)` additional experiments on the latest models and Anyres models, `(2)` a more thorough discussion of visual sink tokens with analysis on large-scale datasets, and `(3)` experiments on the robustness of hyperparameters. This feedback has significantly improved our paper, and we are pleased to note that two reviewers have increased their scores from 5 to 6 and 6 to 8, respectively. We are grateful to all the reviewers for giving us the opportunity to strengthen our work and for finding our research interesting and valuable.

In particular, we had a detailed discussion with Reviewer P74f regarding the interpretation of Figure 3(b). **The primary goal of this experiment is to demonstrate that the visual sink token has a negligible impact on the model's performance**. The key point is that the model remains stable even when the sink token is removed during the inference process, indicating that it does not play a significant role in generating image-related responses. Specifically, this is explained by **the stable performance of the model in the range of $\tau = 20$ to $\tau = 50$, as shown by the red line in Figure 3(b)**. Additionally, through our discussions with the reviewers, we have provided further evidence to support the claim that the visual sink token has a minimal impact on model performance:

1. Removing visual sink tokens does not significantly affect performance, whether *before* or *after* the softmax calculation. (Response to Reviewer P74f (2/3))
2. The final contribution value of the visual sink token $\Vert \alpha ^ {\ell, h} _ {i, j} \boldsymbol{x} _ j ^ {\ell-1} \boldsymbol{W} _ {OV} ^ {\ell, h} \Vert$ is significantly lower than that of other visual tokens. (Further Response to Reviewer P74f (2/3))
3. Visual sink tokens are mainly located in semantically meaningless areas. (Response to Reviewer b7X3 (2/3), Appendix A.2)
4. Visual sink tokens exhibit similar characteristics to text sink tokens, which are known to have a negligible impact on the model (Response to Reviewer KaLi (1/2), Section 4.2, Appendix A.2).

Based on these points, we conclude that the visual sink token has a negligible impact on model performance, which leads us to present Visual Attention Redistribution (VAR) based on this conclusion.

However, in the discussion with Reviewer P74f, a disagreement arose regarding the experiment in Figure 3(b), which **diverged somewhat from the core focus of our experiment**. From our understanding, the reviewer's main concern is why the performance degradation in the "visual sink token" experiment (red line) between $10 < \tau < 20$ is more significant than that in the "random visual token" experiment (blue line) for $\tau > 40$, even though the number of masked tokens is larger in the latter case. Comparing the performance degradation between the two experiments in different ranges of $\tau$ leads to a fuzzy interpretation, as the two experiments are not controlled for the same number of masked tokens, token positions, dynamic vs. fixed masking, and other factors (Further Response to Reviewer P74f (3/3) & Further Response to Reviewer P74f Ⅱ (3/3)). Nevertheless, based on our insights into the visual sink token and related works, we have made our best effort to clarify the interpretation of the experiment.

If we understand the reviewer's response correctly, the main reason for the disagreement is that the reviewer's own experiment results (fixed-position pruning) are not consistent with our experimental results. In the reviewer's experiment, pruning visual tokens did not lead to significant performance degradation, whereas in our experiment, masking resulted in a performance drop. Despite our best efforts, we were unable to replicate the reviewer's results. Furthermore, other studies that conducted masking experiments in similar settings reported significant performance/probability drops with only a few masked tokens [1, 2], which supports our findings.

In this regard, we kindly request that the AC consider the discussion with Reviewer P74f and our responses in the context of the entire paper. The reviewer also mentioned that "*most other aspects of this work appear quite solid*" (Response to Authors (1/2) of Reviewer P74f). We hope that the reviewer's concerns have been addressed in our final response, and we ask the AC to carefully consider the overall aspects of our work.

---

> ### Author Response · Authors · 2024-12-04
> **Final General Response (2/2)**
>
> We believe that the experiments we conducted should not confuse or hinder the reader's understanding of our work. We will provide more intuitive explanations and organize the experimental settings to make it easier for readers to understand the novel phenomenon of the visual sink token and to reproduce the experiments in the final version of the paper.
>
> Finally, we sincerely appreciate the reviewers for their constructive feedback, which has greatly improved the clarity and substance of our work. We are grateful for the opportunity to engage in this discussion and to enhance our research through this process.
>
> Sincerely,
>
> The Authors
>
> [1] Geva, Mor, et al. "Dissecting Recall of Factual Associations in Auto-Regressive Language Models." Proceedings of the 2023 Conference on Empirical Methods in Natural Language Processing. 2023.
>
> [2] Neo, Clement, et al. "Towards Interpreting Visual Information Processing in Vision-Language Models." 2024.

---

> ### Public Comment · ~Haonan_Wang1 · 2024-12-05
>
> Hi,
>
> I very much appreciate your diligence during the author-reviewer discussion period. I have a question regarding your fourth point: "Visual sink tokens exhibit similar characteristics to text sink tokens, which are known to have a negligible impact on the model."
>
> Contrary to this statement, previous works [1][2] indicate that text attention sinks can have a significant impact on model performance. Could you please clarify this point further? I agree that the value vector in LLM attention has a small norm, which aligns with observation on image side. However, there are some differences to consider—for example, removing the text attention before the softmax calculation can have a significant impact on the model's performance.
>
> [1] Ge, Suyu, et al. "A little goes a long way: Efficient long context training and inference with partial contexts." arXiv preprint arXiv:2410.01485 (2024).
>
> [2] Xiao, Guangxuan, et al. "Efficient streaming language models with attention sinks." arXiv preprint arXiv:2309.17453 (2023).

---

> > ### Public Comment · ~Seil_Kang1 · 2025-02-24
> > **RE: Public Comment by Haonan Wang**
> >
> > Dear Hanoan Wang,
> >
> > Thank you for your interest in our work and for providing valuable discussion points. We sincerely appreciate your comments. Apologies for the late response, as public comments are unavailable during the anonymous review period.
> >
> > We acknowledge the importance of attention sinks, as highlighted in the works you mentioned [1][2]. Specifically, we believe their importance can be understood in two aspects.
> >
> > 1. The first aspect concerns their direct influence on the model’s decision-making process. Our experiments indicate that this direct impact is negligible, as the visual sink token is typically located in the background, and its value vector has a small norm. The strong performance of SoftMax-off-by-One in Xiao et al. [2] further supports this observation.
> > 2. The second aspect pertains to the effect of removing the attention sink. Eliminating the attention sink can significantly impact the model’s performance due to its extremely high attention weights. While the visual sink token also exhibits high attention weights relative to other visual tokens, these weights remain lower than those of the `[BOS]` token in text (though the fundamental reason for this remains unclear). Consequently, the r emoval of the visual sink token has a smaller overall effect on the model’s performance compared to text attention sinks.
> >
> > In summary, the visual sink token's first aspect, similar to text attention sinks, has a negligible impact on the model's response due to its minimal direct influence. However, the second aspect—its effect when removed—is less significant for visual sink tokens compared to text attention sinks, owing to their lower attention weights.
> >
> > Thank you once again for your insightful comments.
> >
> > Best regards,
> > The Authors

---

### Meta-Review · Area_Chair_NNmg · 2024-12-20

**Metareview:**

### Paper Summary:
This paper introduces and analyzes the "visual attention sink" phenomenon in Large Multimodal Models (LMMs), where irrelevant visual tokens receive disproportionately high attention weights. The authors propose Visual Attention Redistribution (VAR) to reallocate attention from sink tokens to more relevant visual information, improving model performance without additional training.

### Strengths:
1. Novel phenomenon identification [1YMG]:
> "The identification of visual attention sinks draws parallels with attention sinks in language models, providing a novel insight into the functioning of LMMs."

2. Simple yet effective solution [P74f]:
> "The proposed Visual Attention Redistribution (VAR) method is both innovative and practical. By reallocating attention from sink tokens to relevant visual tokens, VAR improves focus on key image content."

3. Comprehensive validation [b7X3]:
> "The experiments in this paper cover a comprehensive range of content and consistently achieve stable improvements across various tasks."

### Weaknesses:
1. Hyperparameter sensitivity [1YMG]:
> "The reviewer's biggest concern is about the determination of hyperparameter ρ. According to the paper and Fig 7(b), it is benchmark-dependent."

2. Reproducibility concerns [P74f]:
> "Pruning 25% of visual tokens at fixed positions does not lead to significant performance degradation... Could the authors provide more details about their experimental setup?"

### Justification:
While P74f raised important concerns about experimental reproducibility, these focus primarily on one validation approach rather than the core contribution. The phenomenon is well-documented through multiple approaches, and the solution shows practical benefits. As noted by P74f:
> "Most other aspects of this work appear quite solid"

Two other expert reviewers rated the paper above acceptance threshold, acknowledging its contribution to understanding LMM behavior. However, the authors should add detailed experimental protocols and acknowledge limitations in token masking experiments.

As a final remark, the identification of the attention sink phenomenon itself, in my opinion, represents a valuable contribution to the field. A paper can merit acceptance even if readers do not fully agree with the authors' proposed explanation, as the explanation why / how modern deep models work is constantly changing. Since no fundamental flaws were found in the technical work, I recommend acceptance.

**Additional Comments On Reviewer Discussion:**

The key debate centered on token masking experiments (Figure 3b). P74f's experiments showed different results from authors:
> "I randomly selected 25% of the visual tokens and pruned these same fixed positions across all layers. The resulting POPE test score was 82.4"

Authors defended their methodology and added evidence:
> "Pruning involves excluding visual tokens before they enter the layer, while masking is a post-processing step that eliminates their impact after attention calculation"

Despite disagreement on this experiment, the core claims are supported by multiple other validations:
1. Quantitative validation on widely-adopted datasets (Pascal-VOC, MS-COCO) showing sink tokens appear in background regions
2. Consistent improvements across recent models (Qwen2-VL, InternVL2)
3. Analysis showing token contributions align with theoretical predictions

---

### Decision · Program_Chairs · 2025-01-22

Accept (Poster)